# Epidermal development requires ninein for spindle orientation and cortical microtubule organization

Nicolas Lecland*, Chiung-Yueh Hsu*, Cécile Chemin, Andreas Merdes, Christiane Bierkamp

In mammalian skin, ninein localizes to the centrosomes of progenitor cells and relocates to the cell cortex upon differentiation of keratinocytes, where cortical arrays of microtubules are formed. To examine the function of ninein in skin development, we use epidermis-specific and constitutive ninein-knockout mice to demonstrate that ninein is necessary for maintaining regular protein levels of the differentiation markers filaggrin and involucrin, for the formation of desmosomes, for the secretion of lamellar bodies, and for the formation of the epidermal barrier. Ninein-deficient mice are viable but develop a thinner skin with partly impaired epidermal barrier. We propose two underlying mechanisms: first, ninein contributes to spindle orientation during the division of progenitor cells, whereas its absence leads to misoriented cell divisions, altering the pool of progenitor cells. Second, ninein is required for the cortical organization of microtubules in differentiating keratinocytes, and for the cortical re-localization of microtubule-organizing proteins, and may thus affect any mechanisms that depend on localized microtubule-dependent transport.

## Introduction

Microtubules are cytoskeletal polymers, best known for their role during cell division, when they form the spindle apparatus. In interphase, microtubules serve as tracks for intracellular transport of vesicles and other cargoes, and they participate in the remodeling of cell shape during locomotion or during differentiation-specific morphogenesis. Motile cells, such as fibroblasts or lymphocytes, or undifferentiated cells in culture usually show a microtubule network that is radially organized, with the centrosome acting as a microtubule-organizing center. In contrast, many specialized cells in tissues show microtubule arrays that are no longer connected to the centrosome (Dyachuk et al, 2016). An evolutionary conserved protein that has been implicated in the anchorage of microtubules to both centrosomal and non-centrosomal sites is ninein

(Mogensen et al, 2000; Zheng et al, 2016). Ninein possesses an amino-terminal domain that binds to dynein/dynactin (Casenghi et al, 2005), followed by a large coiled-coil–forming central domain. In undifferentiated cells, ninein is bound to the centrosome, and is particularly enriched at the subdistal appendages of the mother centriole and the basal body of the primary cilium, where it binds to microtubule minus-ends (Mogensen et al, 2000; Piel et al, 2000; Delgehyr et al, 2005). Loss of ninein in cultured cells leads to loss of microtubule anchorage at the interphase centrosome, and to multipolar spindles in mitosis (Dammermann & Merdes, 2002; Logarinho et al, 2012). Mutations in the *ninein* gene have been linked to Seckel syndrome, a recessively transmitted human disorder that leads to primordial dwarfism, microcephaly, cognitive defects, and increased sensitivity to genotoxic stress, but the roles of ninein in this pathogenesis are not understood (Dauber et al, 2012). In the developing mammalian neocortex, ninein has been shown to contribute to asymmetric centrosome inheritance, interkinetic movement, and the maintenance of progenitor cells (Wang et al, 2009; Shinohara et al, 2013). In *Drosophila*, a maternally provided *ninein* ortholog, *Bsg25D*, is necessary for proper spindle formation during early embryonic cleavages but is no longer essential at later stages of development (Kowanda et al, 2016; Zheng et al, 2016).

In polarized mammalian epithelial cells that undergo differentiation, ninein re-localizes from the centrosome to non-centrosomal sites in the apical region. This is accompanied by the loss of radial microtubule organization and transformation of the microtubule network towards an apicobasal pattern, with microtubule minus-ends anchored apically and plus-ends extending towards the basal area (Mogensen et al, 2000; Moss et al, 2007; Goldspink et al, 2017). Other non-centrosomal organization patterns are seen in neurons and endothelial cells, where most of the ninein localizes to small cytoplasmic particles (Baird et al, 2004; Matsumoto et al, 2008; Ohama & Hayashi, 2009), and in epidermal microtubule arrays of *Caenorhabditis elegans* and vertebrates, where ninein and ninein homologues localize to the cell periphery (Lechler & Fuchs, 2007; Wang et al, 2015).

In vertebrate epidermis, keratinocytes originate from asymmetric divisions in the basal epidermal layer. The suprabasal

Centre de Biologie du Développement, Centre de Biologie Intégrative, Université Paul Sabatier/CNRS (Centre National de la Recherche Scientifique), Toulouse, France

Correspondence: andreas.merdes@univ-tlse3.fr; christiane.bierkamp@univ-tlse3.fr; leclandn@gmail.com; hsu.misman@gmail.com; cecile.chemin@univ-tlse3.fr
*Nicolas Lecland and Chiung-Yueh Hsu contributed equally to this work

keratinocytes undergo differentiation, during which ninein relocalizes from the centrosome to the cellular cortex. This relocalization is mediated by an interaction between ninein and the desmosomal protein desmoplakin (Lechler & Fuchs, 2007). Besides ninein, the dynein regulators Lis1 and Ndel1, as well as the microtubule plus-end-binding protein CLIP170 also accumulate at the cortex of keratinocytes in a desmoplakin-dependent manner (Sumigray et al, 2011). Concomitantly, microtubules lose their centrosomal anchorage, and a subset of stabilized microtubules aligns with the cortex (Lechler & Fuchs, 2007; Sumigray et al, 2011, 2012). This reorganization of the microtubule network appears to be of major functional importance for the formation of an intact epidermis because the stabilization of cortical microtubules increases the accumulation of components of tight and adherens junctions (Sumigray et al, 2011, 2012). During skin development, large numbers of adherens junctions as well as desmosomes assemble at the entire surface of suprabasal cells. As additional layers of cells are produced from the basal layer, older, more apical cells terminally differentiate to form the spinous and granular layer, where tight junctions are assembled. In the outermost layer, dead cells finally constitute the cornified envelope (CE), containing highly cross-linked proteins and lipids that seal the epidermis. The entirety of intercellular junctions, together with the CE, contribute to adhesion and mechanical stability and impermeability of the skin (Sumigray & Lechler, 2015). This property is termed the "epidermal barrier" and protects the organism from water loss from the inside and from environmental aggressions, such as pathogens or chemicals, from the outside. Interestingly, the integrity of the barrier can be compromised by pharmacological destabilization of microtubules or by knockout of the microtubule-organizing regulator of dynein, Lis1, in the epidermis of mice (Sumigray et al, 2011, 2012; Hsu et al, 2018). The observed barrier defects may be explained in part by defects in tight junctions and desmosomes. Besides, a recent study reported the epidermis-specific loss of microtubules in a subset of keratinocytes in transgenic mice, by tissue-specific overexpression of the microtubule-severing enzyme spastin (Muroyama & Lechler, 2017). In these mice, defects were identified in keratinocyte shape, in epidermal homeostasis, and in desmosome assembly, although the skin barrier was still functional. The interpretation of the results from both Lis1-knockout mice and spastin-mice suffer from various ambiguities: Lis1-dependent defects might in part be due to a pleiotropic effect of the knockout because Lis1 is involved in the regulation of ubiquitous dynein-dependent transport, besides cortical microtubule organization (Sumigray et al, 2011; Egan et al, 2012), and defects in spastin-overexpressing epidermis are difficult to explain because of the mosaic expression pattern in the transgenic mice (Muroyama & Lechler, 2017).

To understand the mechanisms of microtubule reorganization in the developing epidermis, and to clarify the role of ninein, we studied the consequences of ninein knockout in keratinocytes. Here, we present evidence that ninein is essential early in progenitor cells for oriented cell divisions and later in differentiating epidermal cells for the formation of non-centrosomal cortical microtubule arrays. Genetic loss of ninein affects progenitor cell renewal and keratinocyte differentiation, in particular, desmosome assembly and barrier formation.

## Results

### Generation of ninein-deficient mice

To address the role of ninein in epidermal cell differentiation and morphogenesis, we generated two knockout mouse models for *ninein*, driven by Cre-mediated loxP recombination under the control of 1) the *K14* promoter and 2) the *PGK1* promoter. *K14*-Cre induced recombination and excision of ninein exon 2 specifically in basal progenitors of the epidermis from embryonic day (E) 14.5 onwards (Indra et al, 2000). This led to transcripts with altered reading frames, inducing premature stops, and, therefore, generated an epidermal-specific deletion of ninein that we refer to as "cKO" (Fig 1A–C). The second model, using *PGK1*-Cre, directed recombination in all tissues, including the germline, and therefore generated a ubiquitous deletion of ninein that we call "KO" (Fig 1C; Lallemand et al, 1998).

We first generated mice carrying the floxed *ninein* allele Fl/WT and bred them to homozygosity. Homozygous Fl/Fl mice appeared normal, showing that the genetically modified ninein allele was functional. Next, these mice were crossed with *K14*-Cre, and the heterozygous offspring Fl/WT; *K14*-Cre/0 was mated again with homozygous Fl/Fl mice, to obtain "cKO" mice (Fl/Fl; *K14*-Cre/0) and control "WT" animals (Fl/WT; 0/0, Fig 1C, and Fl/Fl; 0/0). The pups of 12 matings (n = 92 animals) were genotyped by PCR and showed that cKO mice were born with the expected Mendelian frequency, indicating that epidermal deletion of ninein was not prenatally lethal. Moreover, cKO mice were viable for at least one year and did not develop any obvious pathology, except for a skin phenotype that will be detailed below.

In our second *ninein*-KO model, Fl/Fl mice were crossed with *PGK1*-Cre mice, and the offspring was obtained in which all tissues, including germline, were heterozygous for the recombined (deleted) ninein allele Del/WT. These mice were crossed "inter se" to obtain ubiquitous ninein-deleted Del/Del animals, which constituted our "KO" animals (Fig 1C). Constitutive deficiency of ninein resulted in 65% viable KO animals that displayed the same skin phenotype as cKO animals (see below) and in 35% lethality (20% prenatal, 15% postnatal, n = 212 Del/Del animals and 225 control animals). The potential role of ninein in these viability-related processes will be addressed in a separate study.

Expected Cre-mediated excision of *ninein* exon 2 was controlled by PCR analysis of the ninein allele in DNA extracts from cKO and KO animal biopsies (Fig 1C). The *ninein* gene produces several widely expressed wild-type transcripts, all of which are protein coding (Bouckson-Castaing et al, 1996; Lin et al, 2006; Zhang et al, 2016). After excision of the floxed region, four transcripts were predicted to produce truncated protein products that may be subjected to nonsense-mediated decay (EUCOM IKMC Project/70308). Analysis of ninein mRNA levels was performed by quantitative real-time PCR in keratinocytes from KO neonates, as well as in fibroblasts from KO embryos, and confirmed efficient deletion of *ninein* (Figs 1D, E, and S1A). We also analyzed whether any protein-coding transcripts were

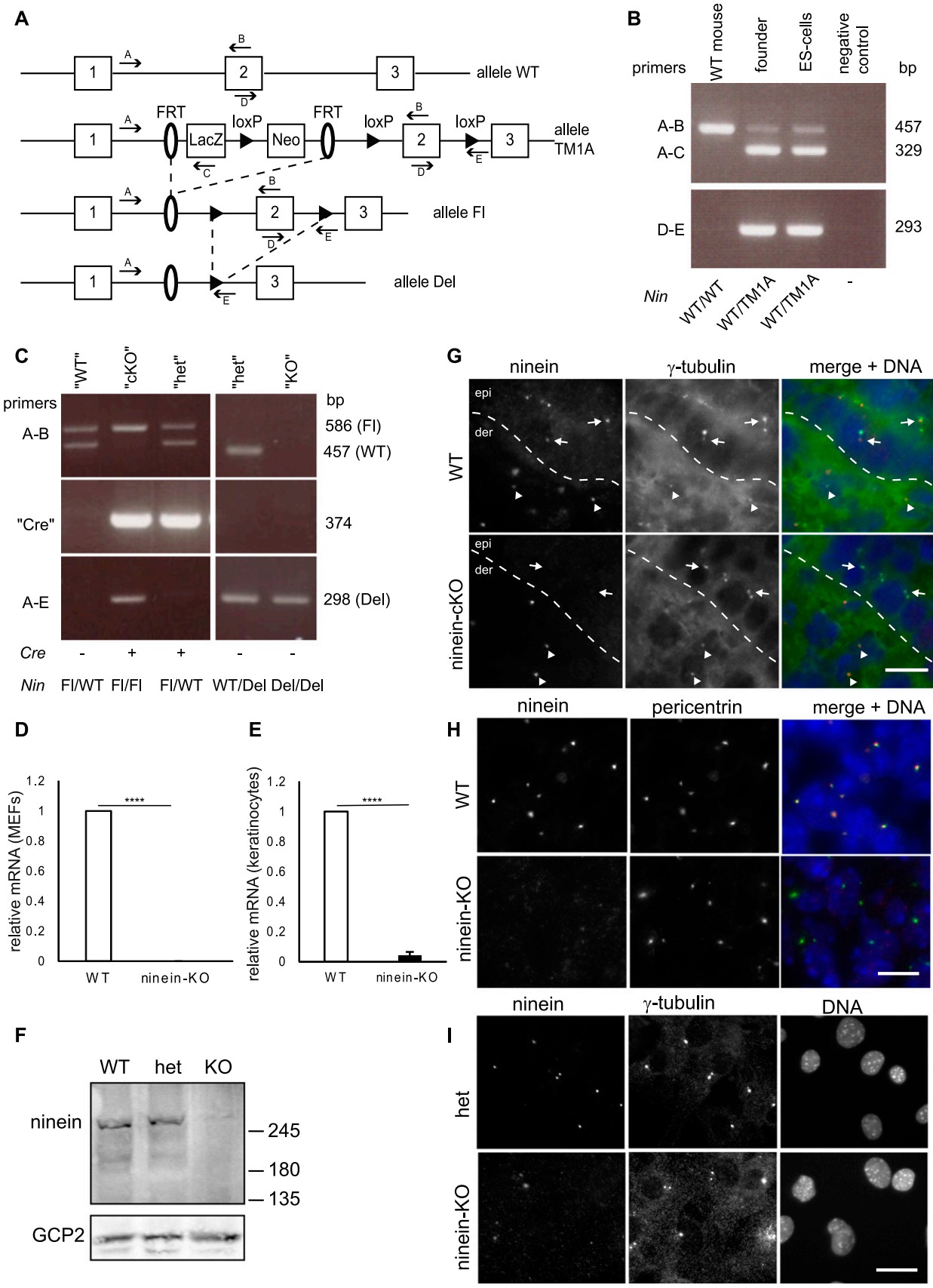

produced upon alternative splicing between exons 1 and 4, but these transcripts were barely detectable in WT or KO cells (Fig S1A).

We verified that excision of *ninein* exon 2 resulted in loss of ninein protein, by immunoblotting of protein extracts from KO embryonic fibroblasts, using an antibody against the C terminus of ninein (Fig 1F). Interestingly, heterozygous cells (het) produced ninein protein levels that were close to those of WT cells (94% compared with WT), suggesting a mechanism of dose compensation (Fig 1F). The absence of ninein protein was further confirmed by immunostaining of frozen skin sections from cKO and KO embryos, using antibodies directed against the N- and C-terminal domains of ninein, and centrosomal proteins γ-tubulin and pericentrin. Epidermal cells of cKO embryos and all cells from KO embryos lacked ninein at the centrosomes but retained γ-tubulin or pericentrin (Fig 1G–I). However, cells of the dermis of cKO, in which K14-Cre was not expressed, showed centrosomal ninein, as was the case for the totality of cells in WT controls (Fig 1G). Finally, we tested whether splicing between exons 1 and 3, after excision of exon 2, may lead to translation of a truncated N-terminal polypeptide of 97 amino acids with a dominant-negative function. Because the first 61 amino acids of such a polypeptide would correspond to the N terminus of ninein, overlapping with a domain for which an interaction with microtubules and dynactin has been shown (Bouckson-Castaing et al, 1996; Casenghi et al, 2005; Delgehyr et al, 2005), we verified that overexpression of a GFP-tagged form of ninein 1–61 in cultured cells had no effect on the localization of endogenous ninein or on the assembly of centrosomes and microtubules (Fig S1B). We can, therefore, assume that the phenotypes resulting from ninein knockout are due to the loss of ninein function.

### Mice with epidermis-specific or ubiquitous loss of ninein show a thinner skin and reduced amounts of epidermal differentiation markers

To study epidermal morphogenesis and homeostasis in *ninein* knockout, we examined cKO and KO newborn pups at the macroscopic level. At first glance, cKO and KO pups were indistinguishable from their control littermates (WT/WT, FL/FL, FL/WT, or Del/WT) and showed no obvious abnormalities of the skin (Fig S2A). However, histological analysis of hematoxylin and eosin (H&E)–stained paraffin sections showed that epidermis from cKO and KO newborns was significantly thinner than that from WT newborns (Fig 2A and B). The most obvious differences were seen in the granular layer that typically displays large keratinocytes in WT skin, containing numerous large keratohyalin granules stained by H&E. By contrast, the granular layer in cKO and KO skin appeared thinner, with cells that had strikingly less and smaller keratohyalin granules (Fig 2A and B). Measurements revealed that the overall relative thickness of cKO and KO epidermis was 67% of WT control epidermis, with the granular layer being more affected (cKO: 49% and KO: 41% of the thickness of WT), whereas basal and spinous layers showed milder effects (cKO: 74% and KO: 87% of the thickness of WT; Fig 2B). Similar results were obtained from electron microscopy of the granular layer (data not shown). Nevertheless, the number of suprabasal cell layers was comparable in the epidermis of WT and ninein-KO mice, counting an average of six cells between the basal layer and the cornified layer (WT, n = 58; ninein-KO, n = 70, from three individuals/each).

To test whether self-renewal or differentiation of keratinocytes was affected by the loss of ninein, we stained newborn skin cryosections with antibodies against keratin 5, as a marker of self-renewing basal layer cells, and with antibodies against keratin 10, as markers of differentiated keratinocytes. However, these markers were correctly expressed in the appropriate layers of cKO and KO neonate skin (Fig 2C). We also analyzed skin sections for the expression of keratin 6 (K6), which is expressed in disease and when homeostasis is perturbed, but we did not observe any significant increase in K6-expressing cells in cKO or KO newborn skin (data not shown).

To examine for potential defects in epidermal differentiation, we examined the presence of markers of the granular layer after loss of ninein function. Profilaggrin is a precursor protein, stored in

---

**Figure 1. Generation of epidermis-specific and constitutive ninein-knockout mice.**
**(A)** From top to bottom: parental genomic locus with exons 1–3 (allele WT); targeted locus containing both *lacZ* reporter and *neomycin* selection genes, flanked by FRT-sites (allele TM1A); the floxed allele had the exon 2 flanked with two loxP sites (allele *Fl*) and the deleted ninein allele upon Cre-mediated excision of exon 2 (allele Del). Boxes represent exons; ovals and arrowheads represent FRT-sites and loxP sites, respectively. **(A–E)** Primers for PCR genotyping (A–E) are indicated as arrows. Upon Cre-mediated deletion, only exon 1 is preserved, which codes for the N-terminal 61 amino acids of ninein. **(B)** PCR analysis of genomic DNA from nontransgenic wild-type mice (WT), transgenic founder mice (WT/TM1A), and ES-cells (WT/TM1A), using primers A, B to amplify the sequence of the WT allele (457 bp) and using primers A, C and D, E for the targeted locus (TM1A) (329 bp and 293 bp), respectively. **(C)** Left: To generate mice with epidermis-specific *ninein* knockout, crosses were made between mice carrying the floxed allele (Fl/Fl), and mice heterozygous for the floxed allele and *K14*-Cre (Fl/WT; *K14* Cre/0). Three different genotypes are characterized by PCR analysis of genomic DNA [tail biopsies], (Fl/WT; 0/0), (Fl/Fl; *K14* Cre/0), and (Fl/WT; *K14* Cre/0). For simplification, they are referred to as "WT," cKO," and "het," respectively. Primers A, B were used to detect the wild-type (457 bp) or the floxed (586 bp) *ninein* allele. Primers A, E detected exclusively the deleted allele (298 bp). Specific primers "Cre" amplified the *K14-Cre* gene (374 bp). Right: for the generation of the constitutive ninein-KO, we crossed (Fl/WT; *PGK1*-Cre/0) mice carrying *PGK1*-Cre, with (Fl/Fl) mice, to generate offspring that had the above characterized, and deleted the ninein allele in all cells and that transmitted it through the germline. *PGK1*-Cre was lost in repeated crossings, yielding heterozygous (het) mice (WT/Del; 0/0) and KO mice (Del/Del; 0/0). **(D, E)** Relative amounts of ninein mRNA in MEFs from WT and *ninein*-KO embryos, or (E) from keratinocytes obtained from WT and *ninein*-KO embryos, determined by quantitative PCR. The value for WT cells was set to 1 (error bars, SEM; triplicates from three independent experiments, ****$P$ < 0.0001). **(F)** MEFs from WT, heterozygous, and *ninein*-KO E14.5 embryos were lysed and probed by Western blotting with antibodies against the C terminus of ninein, and against GCP2 as a loading control. Positions of molecular weight markers (kD) are indicated. **(G)** Immunofluorescence of ninein (N-terminal domain) and γ-tubulin in sections of epidermis from WT or ninein-cKO newborn mice. The dotted line represents the interface between dermis (der) and epidermis (epi); arrowheads and arrows point to centrioles in the dermis or epidermis, respectively. In the WT epidermis, only one centriole within a γ-tubulin–positive pair is ninein positive. Note the absence of ninein from centrosomes in ninein-cKO epidermis. **(H)** Immunofluorescence of tissue (cross-section) from WT and *ninein*-KO E14.5 embryos, using antibodies against ninein (N-terminal domain) and pericentrin. Note the absence of ninein from pericentrin-positive centrosomes in *ninein*-KO tissue. **(I)** Immunofluorescence of MEFs from Het (Del/WT) and *ninein*-KO E14.5 embryos, using antibodies against ninein (N-terminal domain) and γ-tubulin. Some areas of the cytoplasm display unspecific staining with the ninein antibody. Note the absence of ninein from γ-tubulin–positive centrosomes in ninein-KO MEFs. Bars (G, H, I), 10 μm.

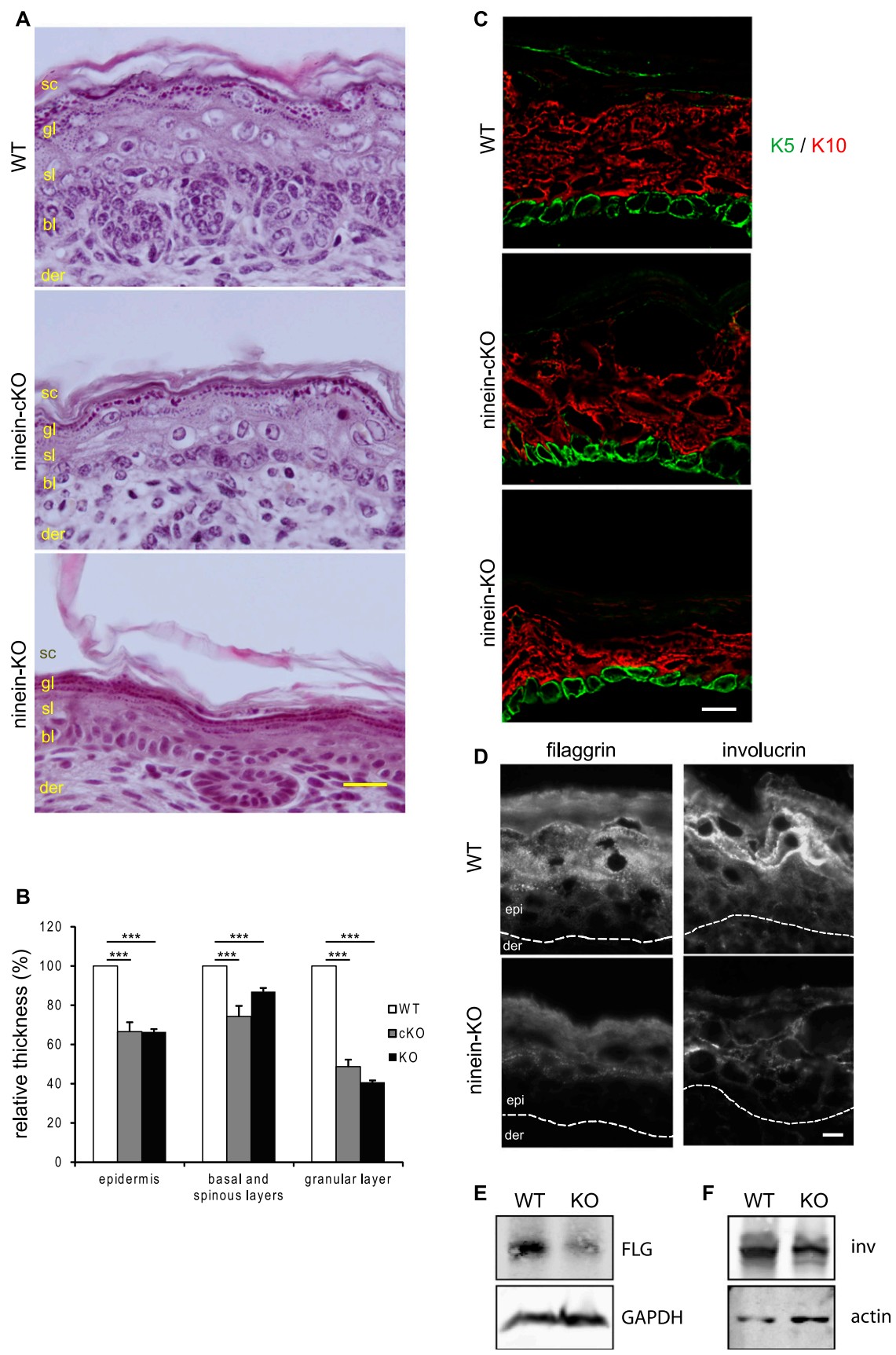

keratohyalin granules in the granular layer, and proteolysed into filaggrin monomers during cornification. We found cytoplasmic localization of (pro)filaggrin in cells of the granular layer and in the stratum corneum of the WT skin (Figs 2D and S2B). The accumulation of (pro)filaggrin in cKO and KO skin was reduced, both in the granular and stratified layers. The low protein levels of filaggrin were confirmed by immunoblotting of protein extracts from KO keratinocytes (Fig 2E). Involucrin, a component of the CE and marker of terminal differentiation of keratinocytes, was localized to the cellular cortex in the cells of the upper granular layer and the stratum corneum of WT skin but was found at much lower levels in the cells of *ninein*-KO skin, and lower protein levels were confirmed by immunoblotting (Fig 2D and F). By contrast, the amounts of the CE marker loricrin remained unchanged in the *ninein*-KO epidermis (Fig S2C). To test whether the reduction of various epidermal markers was due to lowered transcription levels, we quantified the relative amounts of mRNA of filaggrin, desmoglein 1, corneodesmosin, and ninein-like protein in the skin from neonate WT and ninein-KO mice (Fig S2D). Because mRNA levels remained largely unaffected, we assume that the loss of epidermal markers in ninein-KO epidermis was due to altered protein stability. Altogether, our results, thus, show that loss of ninein affects epidermal morphogenesis, generating a thinner skin.

### Increased numbers of EdU-positive cells in the suprabasal compartment and defects in mitotic spindle orientation in the absence of ninein

To identify the cellular mechanisms leading to a thinner skin upon ninein deficiency, we determined whether proliferation was affected in the epidermis after the onset of stratification. We injected embryos (E16.5) with EdU and determined the number of cells in the S-phase (EdU-positive) in the basal (K5-positive) and suprabasal (K10-positive) compartments (Fig 3A). We observed twice as many EdU-positive cells in the suprabasal layer of ninein-KO embryos (15.2% in *ninein*-KO compared with 7.7% in WT embryos, *P* < 0.001), concomitant with a slight reduction in the proportion of EdU-positive cells in the basal layer of ninein-deficient embryos (28.4% in *ninein*-KO compared with 31.8% in WT embryos) (Fig 3B). Interestingly, suprabasal EdU-positive cells were negative for K5, but expressed K10, suggesting that these cells had entered differentiation (Figs 2C and 3A). These results were confirmed by staining E16.5 skin sections with anti-Ki67, an antibody that labels cycling cells but not the cells in the G0 state. Consistently, twice as many Ki67-positive cells were observed in the suprabasal layers of *ninein*-KO mice (20%), compared with control mice (10%; Fig S3A and B). Thus, a higher percentage of suprabasal cells in *ninein*-KO skin fail to enter a G0 state.

To determine whether *ninein*-KO interfered with the cell cycle profile, we analyzed epidermal cells from mutant embryos (E17.5) for DNA content and for mitotic markers. However, neither flow cytometry nor the analysis of mitotic cells (phospho-histone H3–positive) revealed significant differences (Fig S3C–E; 0.5% H3P+ cells in *ninein*-KO, compared with 1% in control cells). In addition, there was no evidence for aneuploidy or polyploidy in *ninein*-KO keratinocytes, and the percentage of caspase 3–positive cells, determined by immunofluorescence and flow cytometry, was not increased in mutant epidermis (0.6% in *ninein*-KO cells, compared with 1% in control cells), suggesting that loss of ninein did not increase apoptosis (Fig S3E).

Because we observed no obvious differences in the cell cycle profile of *ninein*-KO keratinocytes, we investigated other mechanisms related to epidermal differentiation: it has been well documented that at the onset of epidermal stratification, cell divisions shift from predominantly parallel/symmetric to perpendicular/asymmetric, thus generating suprabasal cells that enter differentiation. Spindle orientation depends on astral microtubules interacting with the cell cortex. Because the minus-ends of astral microtubules need to be anchored to the poles and ninein is needed for pole integrity (Logarinho et al, 2012), we tested whether ninein deficiency affected astral microtubules and spindle orientation in any way. We first performed RNA silencing of ninein (knockdown, KD) in mouse primary epidermal progenitor keratinocytes (MPEK), where ninein is normally concentrated at the centrosome, in addition to diffuse localization in the cytoplasm. With 73% transfection efficiency of siRNA in MPEK cultures, we reduced the protein levels of ninein to approximately 11%, as measured by photometric analysis. Upon siRNA treatment, MPEK in mitosis formed bipolar spindles, but 84% of these spindles had almost no astral microtubules, compared with 13% in control MPEK (n = 62 and 67 spindles, analyzed in controls and Nin-siRNA conditions, in ≥3 independent experiments; *P* = 0.001; Fig 3C). Both, length and average fluorescence intensity of astral MTs significantly diminished in ninein knockdowns, to 31 and 21% of WT values, respectively (*P* < 0.001; Fig 3D). Moreover, the division angles in late metaphase were different: whereas control MPEK cells (n = 59) had their mitotic spindles oriented parallel to the substratum (angle between the substratum and the longitudinal axis connecting both spindle poles <30°C), spindles in ninein knockdown MPEK cells (n = 62) showed a striking reduction of planar orientation (69% in KD compared with 100% in control cells), with an important increase in oblique (>30°) or randomly oriented spindles (31% in KD compared with 0% in control cells, *P* < 0.001; Fig 3E). Similar results were obtained when spindle orientation was analyzed in HeLa cells depleted of ninein (n = 60 control RNA-treated cells; n = 50 ninein siRNA-treated cells), confirming a role of ninein in spindle orientation

---

**Figure 2. Mice lacking ninein grow a thinner epidermis with impaired differentiation of suprabasal cells.**
**(A)** Sections of the epidermis from WT or *ninein*-cKO newborn mice were stained with hematoxylin and eosin. Sc, stratum corneum; gl, granular layer; sl, spinous layer; bl, basal layer; der, dermis. **(A, B)** Relative thickness of epidermal layers, as shown in (A); values for the WT were set to 100%. Error bars, SEM; at least four skin sections (three regions per section) from at least four embryos (three independent matings) were measured; ***P < 0.001. **(C)** Sections of epidermis from WT, *ninein*-cKO, and *ninein*-KO newborn mice were stained with antibodies against keratin 5 (green) and keratin 10 (red). **(D)** Sections of the epidermis from WT or ninein-KO newborn mice were stained with antibodies against filaggrin or involucrin, as indicated. The dotted line represents the interface between dermis (der) and epidermis (epi). **(E, F)** Western blots of protein extracts from the epidermis of WT or ninein-KO embryos, probed with antibodies against (E) filaggrin and GAPDH (loading control) or with antibodies against (F) involucrin and actin (loading control). Bars (A), 20 μm; (C, D) 10 μm.

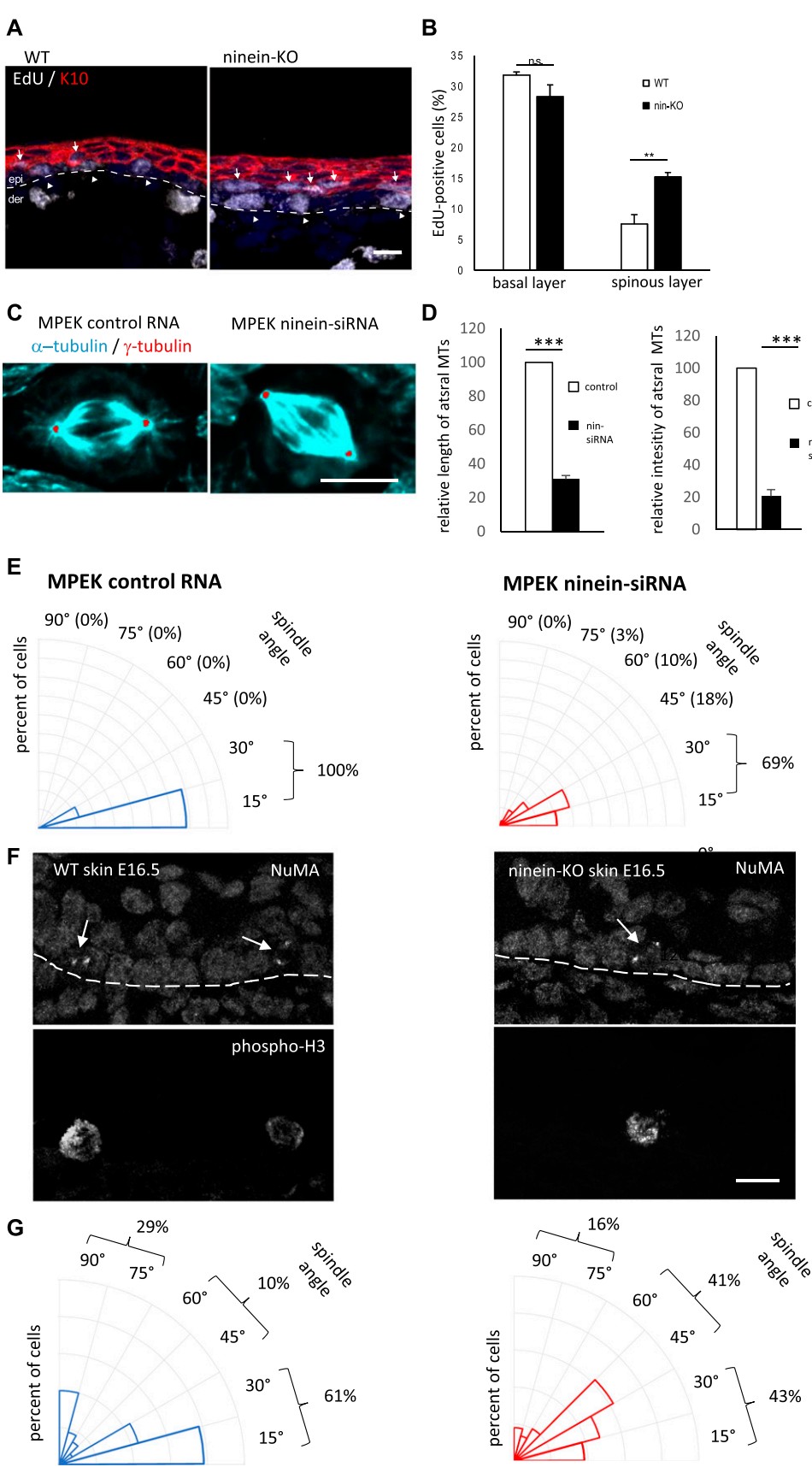

**Figure 3.  Defects in spindle orientation in *ninein*-KO epidermis correlate with suprabasal differentiation defects.**
**(A)** Sections of the epidermis from WT or *ninein*-KO E16.5 embryos after 1 h EdU incorporation were stained with antibodies against keratin K10 and EdU, to detect cells in the S-phase. The dotted line represents the interface between dermis and epidermis; arrowheads and arrows point to EdU-positive cells in the basal and suprabasal layers, respectively. **(B)** EdU-positive cells, as shown in (A), were counted in images covering at least 100 *μ*m of basal and suprabasal layers (error bars, SEM; sections were quantified from three different embryos, five sections per embryo; **P < 0.001). **(C)** Metaphase spindles of keratinocytes (MPEK) after treatment with control RNA or with ninein-siRNA and stained with antibodies against α-tubulin (green) and γ-tubulin (red). For better visualization, the contrast of γ-tubulin was enhanced, the corresponding area selected with the tracing tool and overlaid onto the α-tubulin image. The staining of α-tubulin is slightly overexposed, to better visualize the astral microtubules. **(D)** Astral microtubules in mitotic MPEK were quantified for (left) length and (right) average intensity between spindle poles and the cell cortex, in cells treated with control or ninein-siRNA (n = 20 spindles per condition in at least three independent experiments, ***P < 0,001). **(E)** Radial histograms representing the frequency of spindle orientation angles in (left) dividing control and (right) ninein-siRNA–treated keratinocytes. Spindle orientation was determined by measuring the angle between the line connecting both spindle poles and the surface of the coverslip. 59 and 62 spindles in metaphase were quantified after control RNA and ninein-siRNA, respectively, in three independent experiments, ***P < 0.001. **(F)** Sections of the epidermis from WT or ninein-KO E16.5 embryos, stained with antibodies against NuMA and H3P, to detect spindle poles of mitotic basal progenitors. The dotted line represents the separation between dermis and epidermis and corresponds to the basal membrane. **(G)** Radial histograms representing the frequency of spindle orientation angles in (left) dividing control and (right) *ninein*-KO basal keratinocytes in the epidermis. Spindle orientation was determined by measuring the angle between the line connecting both spindle poles and the basal membrane. Sections were quantified from three different embryos, five sections per embryo; 46 to 58 metaphase spindles were quantified per genotype *P < 0.05). Bars (A), 20 *μ*m; (C, F) 10 *μ*m.

in different epithelial cell models (Fig S3F). We then measured the division angle of mitotic basal cells in sections of epidermis (E16.5), stained for phospho-histone H3 and NuMA (Nuclear/Mitotic Apparatus protein) (Fig 3F–G). Although the exact phase of mitosis was not determined with these two markers, a notable difference in the distribution of spindle angles was seen between epidermis from controls and *ninein*-KO: in control epidermis, most spindles (n = 46) oriented in a bimodal fashion (Williams et al, 2011), either symmetrically with an angle parallel to the basal layer between 0° and 30°, or asymmetrically with an angle between 60° and 90° (Fig 3G). However, consistent with our results on siRNA-treated MPEK cultures, we observed an important increase in oblique spindles in *ninein*-KO epidermis (n = 58 cells), with angles between 30° and 60° (41% in *ninein*-KO compared with 10% in control mice, *P* < 0.005; Fig 3G), concomitant with a reduction of both planar orientations (43% in *ninein*-KO compared with 61% in control mice) and perpendicular orientations (16% in *ninein*-KO compared with 29% in control mice; Fig 3G). In accord with reduced numbers of planar orientations, histological analysis revealed 18% less basal cells per unit area in the epidermis of newborn *ninein*-KO mice, as compared with WT (at least four skin sections from at least three embryos from two independent matings were measured, *P* < 0.05). Because altered spindle orientation in *ninein*-KO epidermis may influence the cell cycle status, cell signaling, and the differentiation program in suprabasal cells (Williams et al, 2011), we performed immunofluorescence of the Notch3 signaling effector, that is, the intracellular domain NICD3, in suprabasal layers of *ninein*-KO and control epidermis (Fig S4A). Intranuclear NICD3 was seen in both, suggesting that ninein-dependent spindle orientation defects may only have a minor influence on the suprabasal differentiation program or may have an influence only in a minor fraction of cells. Related to this, we tested for the presence of primary cilia in *ninein*-KO epidermis because these are required for epidermal differentiation and homeostasis (Croyle et al, 2011; Ezratty et al, 2011) and diverging views have been published on the importance of ninein in ciliogenesis (Graser et al, 2007; Mazo et al, 2016). Whole-mount immunofluorescence was performed at the onset of stratification (E13.5), labelling ODF2 at subdistal appendages and acetylated tubulin at ciliary microtubules (Fig S4B). Apically oriented primary cilia were easily detectable in the 3D stacks of *ninein*-KO epidermis. The proportion of basal cells containing a primary cilium matched that found in the WT epidermis (80% ± 18% SEM in *ninein*-KO compared with 73% ± 18% SEM in WT epidermis *P* > 0.05; at least 100 cells per embryo were analyzed, and five to six embryos per genotype from two to three independent matings; Fig S4B and C). This demonstrates that differentiation defects in ninein-deficient epidermis are not a consequence of defective ciliogenesis or ciliary signaling.

### Ninein is responsible for cortical microtubule re-organization and cortical anchorage of Lis1 in epidermal cells

Because ninein-dependent alterations in spindle orientation did not alter significantly early differentiation events in the immediate suprabasal layers of the epidermis, such as Notch signaling, formation of primary cilia, or K10 expression, we searched for additional mechanisms that may explain the ninein-dependent

reduction of terminal expression markers filaggrin and involucrin (Fig 2D–F). Because our earlier work has shown that the expression of epidermal differentiation markers depends on stable microtubules (Hsu et al, 2018), we investigated the microtubule network in differentiated keratinocytes deficient of ninein.

Microtubule immunofluorescence was performed in primary keratinocytes from mouse embryos in culture because their visualization in epidermal tissue proved to be difficult technically, providing only limited resolution (Lechler & Fuchs, 2007; Hsu et al, 2018). Differentiation of keratinocytes was induced by raising the concentration of extracellular $Ca^{2+}$, leading to the massive formation of adherens junctions and desmosomes (Hennings et al, 1980). Consistent with published data, an initially radial microtubule cytoskeleton was transformed into an array of cytoplasmic microtubules crossing each other, with high amounts of microtubules enriching at the cell cortex, running parallel to the cell borders in controls (Fig 4A) (Lechler & Fuchs, 2007). By contrast, this cortical microtubule enrichment was lost in *ninein*-KO cells. Concomitantly, the cortical enrichment of Lis1 and of the dynactin subunit p150 was diminished in suprabasal keratinocytes of epidermis from *ninein*-cKO and KO neonate mice (Fig 4B and C). To quantify the loss of cortical microtubule organization, we analyzed MPEK cells treated with ninein siRNA and induced them to differentiate. We plotted the relative *α*-tubulin immunofluorescence intensity as a function of the distance to the cell–cell contacts, labelled by *α*-catenin (Fig 4D and E; see the Materials and Methods section). The resulting graphs revealed two *α*-tubulin peaks, flanking both sides of the *α*-catenin peak, indicating the cortical microtubule arrays near the cell–cell contacts between control MPEK. In ninein-depleted MPEK, these *α*-tubulin peaks were reduced by more than 72% (Fig 4E). Likewise, accumulation of Lis1 at the cortex of differentiating control MPEK (yielding one single peak on the graph and co-localizing with the intercellular contact sites, visualized by *β*-catenin; Fig 4F and G) was completely lost in ninein-depleted MPEK.

### Defects in desmosome assembly and lamellar body secretion in the absence of ninein

Because the assembly and maintenance of desmosomes and the transport of lamellar bodies rely, at least in part, on microtubule-associated proteins and on microtubule-mediated transport (Raymond et al, 2008; Nekrasova et al, 2011; Sumigray et al, 2011), we investigated both structures in the neonate skin of WT, cKO, and KO mice. Immunofluorescence experiments with a collection of antibodies revealed differences of the desmosomal marker desmoglein 1. Compared with the abundant cortical localization in WT suprabasal cells, desmoglein 1 was scarce and strongly reduced at the cortex in cKO and KO suprabasal cells (Fig 5A). The reduction of desmoglein 1 was confirmed by immunoblotting of KO keratinocyte extracts (Fig 5B). In contrast, we did not see alterations of the localization or of the protein amounts of desmoglein 3, plakoglobin, or desmoplakin in mutant skin (Fig S5A). Nevertheless, RNA silencing of ninein in MPEK cells prevented accumulation of desmoplakin at the cortex within a window of 30 min upon $Ca^{2+}$-induced differentiation, resulting in a discontinuous, spotty desmoplakin staining pattern (Fig S5B). This suggests that initial

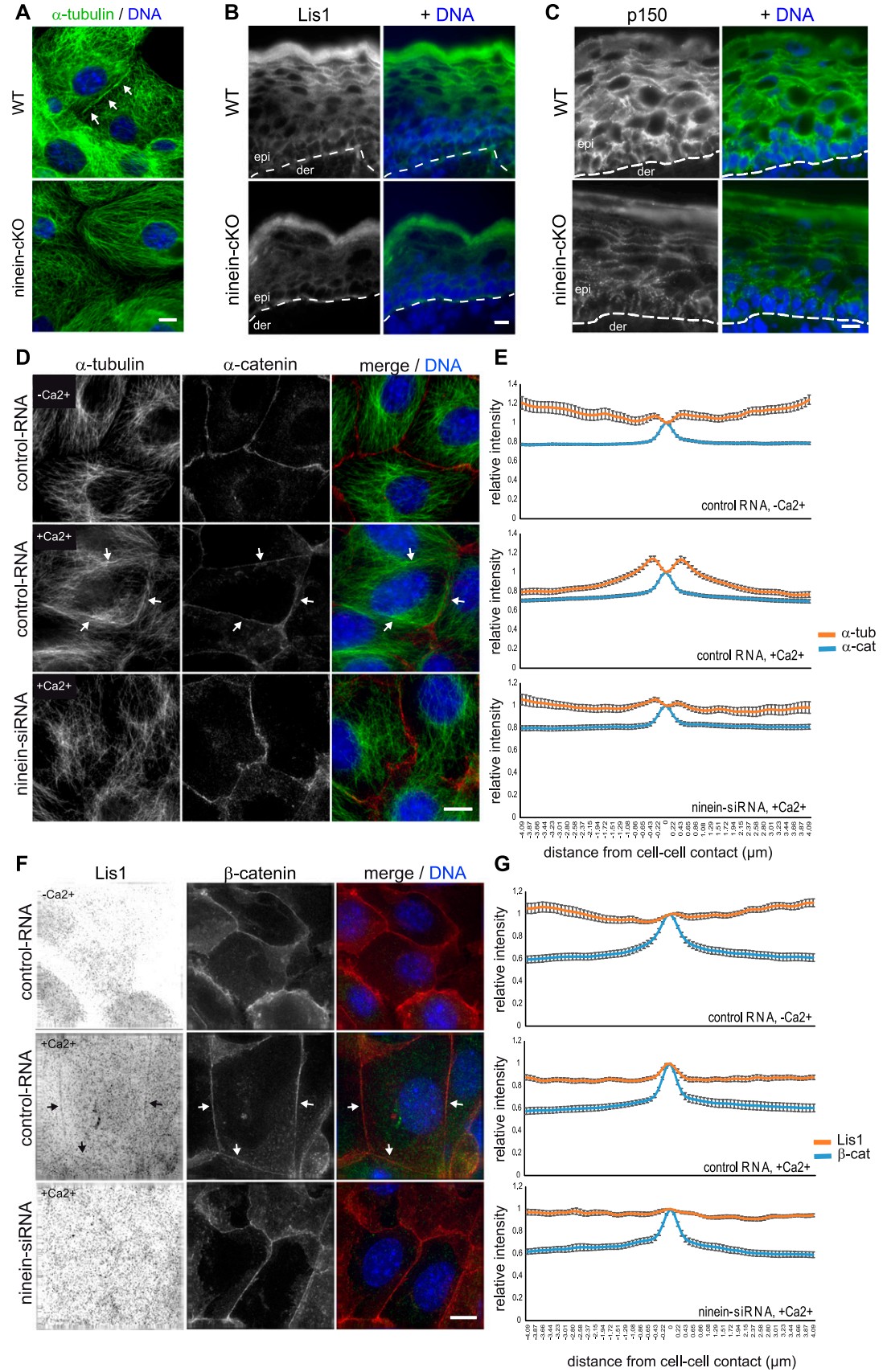

desmoplakin recruitment at the cortex may be slowed down by the absence of ninein, but this effect may be compensated with time. To test whether the structure and integrity of desmosomes was affected by the loss of ninein, we analyzed skin samples from cKO, KO, and WT mice by electron microscopy. High numbers of desmosomes were visible at the intercellular contacts between WT suprabasal keratinocytes (Fig 5C and D). These desmosomes displayed the characteristic morphology with two electron-dense plaques at both sides of the opposing cell membranes, with an extracellular core, and with a desmoglea that appeared as a clear straight line (Fig 5C). Keratins were visible as grey structures at the cytoplasmic surface of each dense plaque (Fig 5C). However, in the skin from ninein-knockout mice, desmosomes were reduced in number (7 and 6 per unit length in ninein cKO and KO, respectively, compared with 13 in WT epidermis), and in size (351 and 328 nm in ninein cKO and KO, respectively, compared with 401 nm in WT; Fig 5D and E). Moreover, we observed qualitative differences in mutant skin, where desmosomes had an irregular morphology, with a faint, kinked desmoglea, and with both plaques appearing less dense, with less keratin filaments attached (Figs 5C and S5D). Although desmosomes were altered in ninein-knockout epidermis, the markers of adherens junctions (β-catenin, Fig S5A, as well as E-cadherin and α-catenin [data not shown]) and of tight junctions (claudin 1, Fig S5A, as well as occludin [data not shown]) all showed correct localization.

In the more apical cornified layer of normal skin, desmosomes are substituted by specialized junctional complexes with a similar morphology, termed corneodesmosomes. One of their components, corneodesmosin, is delivered by secretion from lamellar bodies, a process believed to involve microtubule dependent transport (Ishida-Yamamoto et al, 2004; Raymond et al, 2008). Because lamellar body secretion is essential for epidermal barrier formation (Reynier et al, 2016) and secretory events often involve microtubule-dependent transport, we tested whether any aspect of lamellar body secretion was affected by ninein knockout. As a first step, we stained frozen tissue sections from the WT and KO newborn skin for corneodesmosin. In the WT skin, a very high accumulation of corneodesmosin was observed at the interface between granular layer cells and the stratum corneum, where secretion of lamellar bodies occurs (Fig 6A). In the KO skin, however, this corneodesmosin accumulation was strongly reduced. Remaining immunofluorescence of corneodesmosin was seen in the cytoplasm of granular layer cells, at 53% (+27) intensity compared with granular

layer cells in WT epidermis (n = 40 for each genotype). Correspondingly, slightly reduced protein levels of corneodesmosin were revealed by immunoblotting of tissue extracts of KO skin (67% compared with WT; Fig 6B). We suspected that the major differences of corneodesmosin staining at the stratum corneum were due to reduced secretion of lamellar bodies in ninein-KO epidermis. This was verified by analyzing the lipid deposition onto the plasma membrane of isolated corneocytes, from the WT and KO newborn skin, using the lipophilic dye Nile red. Corneocytes from the KO skin displayed significant weaker staining than their WT counterparts (Fig 6C). To examine directly the localization and morphology of lamellar bodies, we analyzed the WT and KO newborn skin sections by transmission electron microscopy. In the WT skin, we observed many prominent lamellar bodies in the state of fusion with the apical membrane of granular layer cells, facing the stratum corneum (Fig 6D, arrowheads). This was very different in the KO and cKO skin, where only few and small lamellar bodies were found at the apical cell membrane (Figs 6D and S5E). Quantification of the relative secretion area, as defined by the area delimited by the lamellar bodies and the cell membrane over a given distance, confirmed a significant reduction of secretion area in the mutant skin compared with the WT skin (44 and 39% in cKO and KO compared with 100% in the WT epidermis; Fig 6E). That said, we observed comparable amounts of lamellar bodies with intact morphology in the cytoplasm of mutant cells, suggesting that lamellar body synthesis was not affected in mutant cells (Fig 6F and G).

## Mice with epidermis-specific or ubiquitous loss of ninein display defective epidermal barrier formation

Because the qualitative and quantitative molecular and cellular defects in ninein-deficient epidermis are believed to affect cohesion, mechanical resistance, and insulation of the epidermis, we tested whether the epidermal barrier was altered in the ninein mutants. In mouse embryos, the epidermal barrier is acquired towards the end of the embryogenesis, to ensure survival of the neonate in a terrestrial environment at birth (Hardman et al, 1998). We, therefore, examined the outside-in epidermal barrier in WT, cKO, and KO embryos at stages E16.5, E17.5, and E18.5. We subjected them to a permeability assay using the dye X-gal (Hardman et al, 1998). At E16.5, in all WT, cKO, and KO embryos, the stratum corneum was not completely matured and therefore

**Figure 4. Absence of ninein in differentiating keratinocytes prevents cortical organization of microtubules and cortical recruitment of Lis1 and dynactin p150.**
**(A)** Primary keratinocytes, isolated from epidermis of WT and ninein-cKO newborn mice, were grown in culture, differentiated by addition of Ca²⁺, and immunostained with an antibody against α-tubulin. Arrows indicate cortical microtubule arrays in WT cells. Blue, DNA. **(B, C)** Immunofluorescence (green) of Lis1 or (C) dynactin p150 in epidermal sections from WT or ninein-cKO newborn mice. The dotted lines represent the interface between dermis (der) and epidermis (epi). **(D)** Immunofluorescence of α-tubulin (green) and α-catenin (red) in MPEK, transfected with control RNA or ninein-siRNA. Cells were fixed in growing conditions or after induction of differentiation by addition of Ca²⁺ for 24 h. Arrows indicate the cell cortex, where microtubules are reorganized in control cells. Blue, DNA. **(D, E)** Quantification of the fluorescence intensity of α-catenin (blue) and α-tubulin (orange) in the vicinity of the cell/cell interface, in growing or differentiated cells, as prepared in (D). The maximum value for α-catenin was used as a reference for the position of the cell borders, and intensity values were set to 1 at this position (error bars, SEM; at least 30 cells were quantified for each condition, from two independent experiments). **(F)** Immunofluorescence of Lis1 (green) and β-catenin (red) in MPEK, transfected with control RNA or ninein-siRNA. Cells were fixed in growing conditions or after induction of differentiation by addition of Ca²⁺ for 24 h. Arrows indicate areas of the cortex, where Lis1 is enriched in control cells. Inverted contrast is shown for Lis1, to improve visibility. Blue, DNA. **(F, G)** Quantification of the fluorescence intensity of Lis1 (orange) and β-catenin (blue) in the vicinity of the cell/cell interface, in growing or differentiated cells, as prepared in **(F)**. The maximum value for β-catenin was used as a reference for the position of the cell borders, and intensity values were set to 1 at this position (error bars, SEM; at least 30 cells for each condition were analyzed, from two independent experiments). Bars (A–D, F), 10 μm.

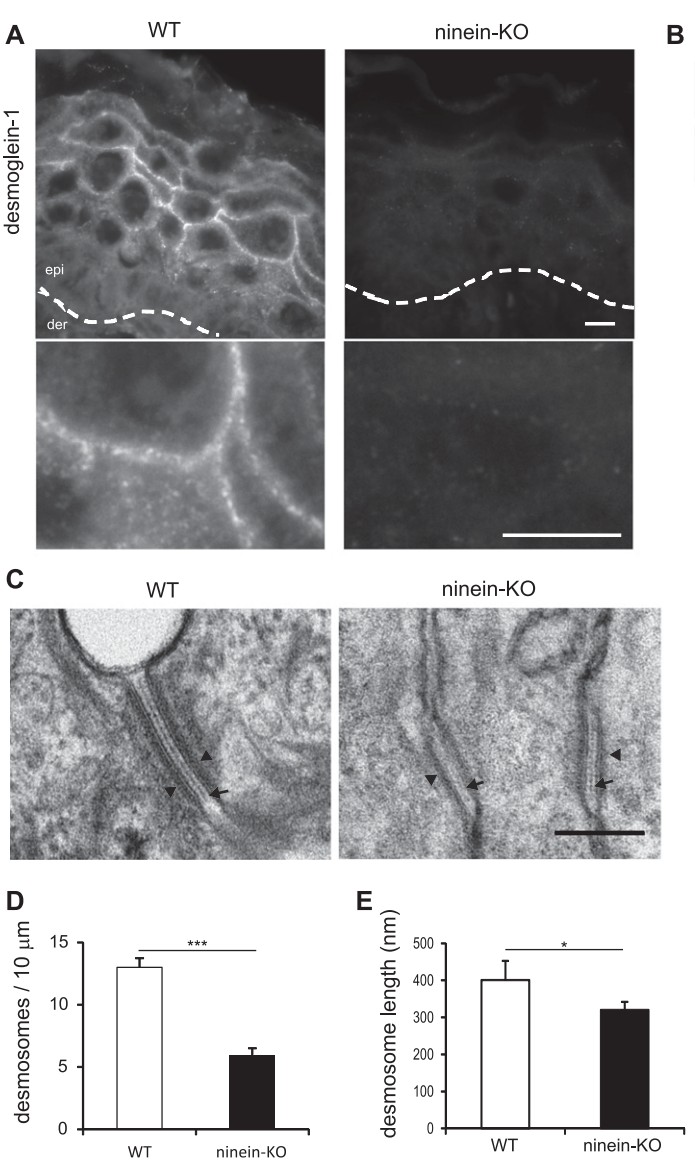

**Figure 5. Lack of ninein reduces the number of desmosomes and alters their structure in the suprabasal epidermis.**

**(A)** Sections of epidermis from WT or *ninein*-KO newborn mice were stained with an antibody against desmoglein-1. The dotted line represents the interface between dermis (der) and epidermis (epi). Enlarged areas below show the continuous immunostaining of desmoglein-1 in WT controls, in contrast to the faint, discontinuous staining in *ninein*-KO epidermis. **(B)** Proteins were extracted from the epidermis of WT or *ninein*-KO newborn mice and probed by Western blotting with antibodies against desmoglein-1 and against actin as a loading control. **(C)** Electron microscopy of desmosomes in the epidermis of WT or *ninein*-KO newborn mice. Cell–cell contacts in the granular layer are shown. The arrowheads and arrows point to attachment plaques and desmosome dense midlines, respectively. **(D)** Number of desmosomes counted along 10 $\mu$m of cell–cell interface of WT and *ninein*-KO newborn suprabasal epidermis (error bars, SEM; at least 10 regions of 10 $\mu$m were analyzed, from four embryos from two independent matings; ****$P <$ 0.0001). **(E)** Desmosome length was measured in WT and *ninein*-KO suprabasal epidermis, as presented in (C) (error bars, SEM; at least 15 desmosomes were measured from four embryos, two independent matings; *$P <$ 0.05). Bars (A), 10 $\mu$m; (C) 200 nm.

permitted dye penetrance, resulting in a blue precipitate in the dermis of the embryos (Figs 7A and S6). One day later, at E17.5, all WT embryos had developed an epidermal barrier on the dorsal part, preventing the penetration of the dye. In contrast, ninein cKO and KO embryos showed dye penetration almost all over their surface (Figs 7A and S6). By E18.5, however, cKO and KO embryos had also developed an epidermal barrier and were impermeable for the dye (Fig 7A). Because the epidermal barrier also protects the organism from dehydration due to inside-out water loss, we performed transepidermal water loss assays (TEWL) on the WT and KO newborn skin. Using an evaporimeter, we measured the water vapour pressure at the skin surface of our neonate mice. The KO skin had significantly higher mean TEWL values (10.11 g/m²h) than the WT skin (8.19 g/m²h, $P$ = 0.01; Fig 7B), suggesting that the inside-out barrier is slightly compromised in KO mice. In summary, absence of ninein resulted in a thinner epidermis with a delayed differentiation and barrier formation.

# Discussion

Using ninein-deficient mice, we demonstrate that ninein is involved in the development of an intact epidermis, by at least two cellular mechanisms: by contributing to spindle orientation during the division of progenitor cells and by organizing a keratinocyte-specific cortical microtubule network.

### Ninein-dependent mitotic spindle orientation in basal keratinocytes

Epidermal architecture and function depends on symmetric, planar divisions, maintaining the pool of basal progenitor cells, and asymmetric, perpendicular divisions. The latter generate a basal progenitor cell capable of self-renewal and a suprabasal cell that differentiates. We present evidence that ninein is

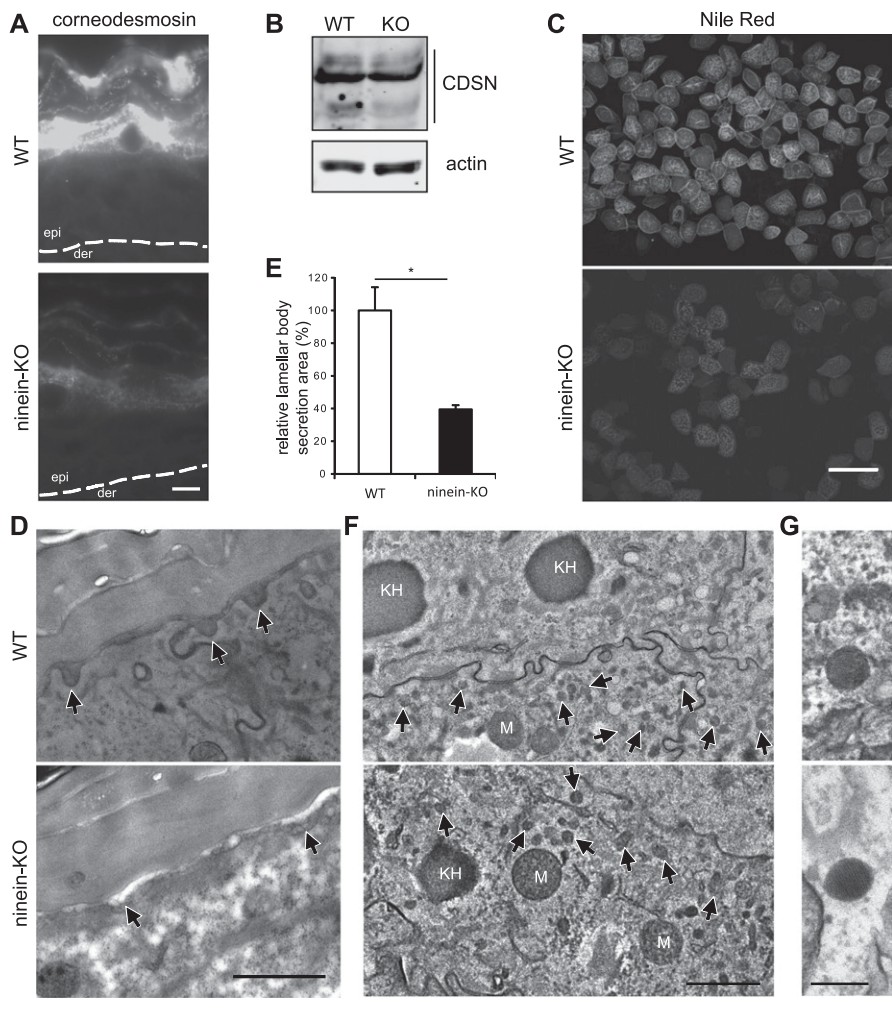

**Figure 6. Lack of ninein impairs secretion of lamellar bodies.**
**(A)** Sections of epidermis from WT or *ninein*-KO newborn mice were stained with an antibody against corneodesmosin. The dotted line represents the interface between dermis (der) and epidermis (epi). **(B)** Proteins were extracted from the epidermis of WT or *ninein*-KO newborn mice, and probed by Western blotting with antibodies against corneodesmosin and against actin as a loading control. Blots were from the same protein extracts as shown in Fig 5B, and the blot against actin is therefore identical. **(C)** Corneocytes from the epidermis of WT and *ninein*-KO newborn mice were isolated and stained with Nile red. **(D)** Electron microscopy of epidermis at the interface between stratum granulosum and stratum corneum, from WT or *ninein*-KO newborn mice. Arrowheads indicate lamellar bodies at sites of secretion. **(E)** Areas of lamellar body secretion, as shown in (D), were outlined and measured. The value for the WT was set to 100% (error bars, SEM; at least 20 secreted lamellar bodies were measured per embryo (n = 4 different embryos, from two independent matings; *P < 0.05). **(F)** Electron microscopy of epidermis from WT or *ninein*-KO newborn mice, showing cytoplasmic lamellar bodies (arrows) in the cells of the lower granular layer. Lamellar bodies can be distinguished by size from the larger mitochondria (M) and keratohyaline granules (KH). **(G)** The lamellar substructure of the lamellar bodies is visible in enlarged views. Bars (A), 10 μm; (C) 100 μm (D, F), 1 μm; (G) 200 nm.

required for proper planar and perpendicular spindle orientation of progenitor cells. Interestingly, ninein has been shown to be part of a multiprotein complex with pericentrin, and loss of pericentrin prevents ninein localization at the mitotic spindle pole, leading to spindle orientation defects (Chen et al, 2014). Moreover, mutations or absence of ninein lead to defects in neural tissue and to microcephaly (Wang et al, 2009; Dauber et al, 2012; Chen et al, 2014). We think that ninein and pericentrin contribute to spindle orientation by anchoring and stabilizing astral microtubules at their minus-ends and, thus, maintain the integrity of spindle poles and centrosomes, in contrast to proteins such as LGN (Leu-Gly-Asn-repeat protein) or NuMA that anchor the plus-ends of astral microtubules to the mitotic cell cortex (Dammermann & Merdes, 2002; Logarinho et al, 2012; Chen et al, 2014). This suggests that anchorage of astral microtubules at both plus- and minus-ends should be necessary for spindle orientation, and defects in either may affect basal cell divisions and epidermal differentiation (Williams et al, 2011). In our *ninein*-knockout mice, spindle misorientation at the beginning of stratification decreases symmetric divisions and as a consequence, reduces the pool of progenitor cells, resulting in 18% reduction of basal layer cells in newborn mice. Overall, this may slow down

the formation of the epidermis and may partly explain the observed delay in terminal differentiation and barrier formation. Because the absence of ninein also affects asymmetric divisions, suprabasal cells may be generated without the full complement of keratinocyte-specific cell fate determinants, thus entering differentiation incorrectly. However, immunostaining of primary cilia, Notch, and K10 in suprabasal cells of *ninein*-KO epidermis indicated that differentiation-specific signaling and expression are largely intact, suggesting that the percentage of keratinocytes in which spindle orientation defects prevent Notch-signaling and differentiation is low (Williams et al, 2011). Consistently, the percentage of suprabasal cells that remain Ki67 positive and thus fail to enter differentiation only increases by approximately 10% (Fig S3B), and the cell cycle profiles of WT and *ninein*-KO keratinocytes show minor differences.

In summary, ninein-dependent defects in cell division may have a modest effect on stratification and epidermal differentiation, because the number of suprabasal cell layers are comparable with the WT control epidermis and early differentiation events remain largely unaffected. We, therefore, searched for additional mechanisms that may explain the developmental defects resulting from ninein knockout.

**A**

WT ninein-KO

E 16.5

E 17.5

E 18.5

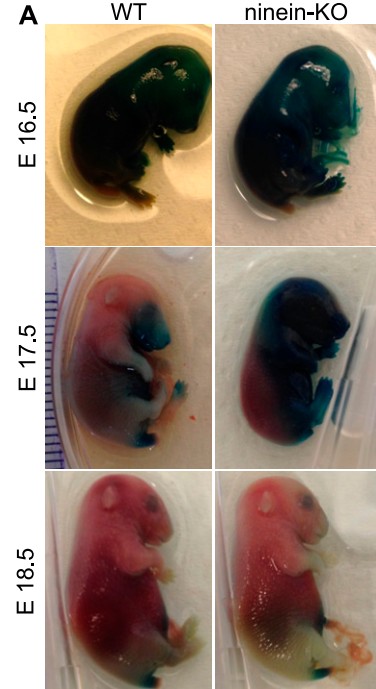

**B**

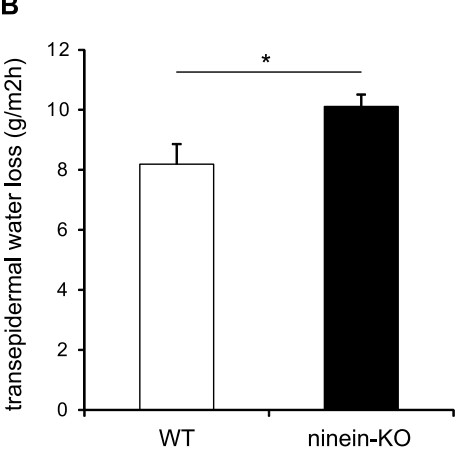

**Figure 7. Lack of ninein results in a defective epidermal barrier.**
**(A)** Barrier assay (X-gal penetration) of WT and *ninein*-KO embryos at different embryonic stages, as indicated. **(B)** Measurements of TEWL were taken from WT and *ninein*-KO newborn mice (error bars, SEM; at least 26 newborns were analyzed, from six independent matings; *$P$ < 0.05).

## Ninein-dependent cortical microtubule organization in suprabasal keratinocytes

The differentiation program of skin keratinocytes involves detachment of microtubules from the centrosome and accumulation of microtubules in the cortical region of the cell. It has been shown that this microtubule reorganization depends on the desmosomal protein desmoplakin and that a variety of microtubule-interacting proteins bind to the cell cortex in a desmoplakin-dependent manner (Lechler & Fuchs, 2007; Sumigray et al, 2011). So far, the mechanism by which desmoplakin exerts its effect on microtubule reorganization has remained unclear. Using two different mouse models in which ninein was knocked out either constitutively or specifically in the epidermis, we provide now direct evidence for an involvement of ninein in the formation of differentiation-specific microtubule arrays at the cortex.

Loss of cortical microtubules after ninein knockout may be due to loss of ninein-dependent microtubule anchorage (Dammermann & Merdes, 2002). Moreover, the absence of ninein may lead to reduced microtubule stability at the cortex because evidence has been provided for a microtubule-stabilizing role of ninein in differentiating neurons (Srivatsa et al, 2015). An earlier view on ninein, as a receptor for γ-tubulin complexes, is not supported by our study because we demonstrate that γ-tubulin is still recruited to the centrosome in ninein-knockout cells (Stillwell et al, 2004; Delgehyr et al, 2005; Lin et al, 2006). This suggests either that earlier reports were documenting indirect effects of ninein on γ-tubulin or that there is redundancy in γ-tubulin anchorage.

Because the amino-terminal domain of ninein interacts with dynein and dynactin (Casenghi et al, 2005), loss of ninein in keratinocytes may also reduce the cortical localization of the dynein-regulating protein Lis1 and of the dynactin subunit p150, as seen in our experiments in cell culture and in vivo. Because loss of Lis1 from the cortex is known to prevent keratinocyte-specific reorganization of microtubules (Sumigray et al, 2011), the effect of *ninein*-KO on the microtubule network may be indirect via Lis1. Furthermore, the ninein-dependent loss of cortical microtubule attachments may interfere with microtubule-dependent transport of any cargoes destined to the cellular cortex.

## Epidermal defects dependent on altered microtubule organization and transport

Two types of cargo that contribute to epidermal function may have been affected in *ninein*-knockout mice: lamellar bodies and desmosomal proteins. Lamellar bodies are lysosome-related organelles that emerge first in keratinocytes of the upper spinous layer. They contain specific lipids and proteins that are secreted after further maturation at the apical cell membrane of the granular layer. The contents of lamellar bodies serve as precursors for the formation of the CE. Because biochemical preparations of lamellar bodies led to the identification of multiple binding partners that mediate microtubule-dependent transport, such as proteins of the kinesin family, components of the dynein/dynactin complex, and CLIP170, it is imaginable that the ninein-dependent organization of microtubules along the cellular cortex supports localized transport and secretion of lamellar bodies (Raymond et al, 2008). Consistently, the small GTPase Rab11a that controls secretory pathways and that is transported along the microtubules by dynein has been found essential for lamellar body biogenesis (Horgan et al, 2010;

Reynier et al, 2016). Any interference with microtubule-dependent transport and secretion may explain our observation of reduced corneodesmosin localization at the interface between the granular and cornified layer because corneodesmosin is a prominent cargo of lamellar bodies (Ishida-Yamamoto et al, 2004). Likewise, reduced secretion of lamellar bodies can explain our data on lowered amounts of lipids in corneocytes. Besides lamellar bodies, another CE precursor, S100A11, also uses microtubules for cortical redistribution (Broome & Eckert, 2004).

In a similar manner, the cortical delivery of several desmosomal proteins is believed to involve microtubule-dependent transport: Nekrasova et al (2011) have shown that the desmosomal cadherins desmoglein 2 and desmocollin 2 require microtubule-dependent transport by motor proteins of the kinesin family, for their rapid accumulation at intercellular junctions. In our experiments, we observe reduced amounts of desmoplakin that are unevenly localized at the cell cortex in the absence of ninein, shortly after induction of differentiation in cell cultures (30 min, Fig S5B) but not after an extended period of time in fully differentiated epidermis in vivo (Fig S5A). This suggests that the kinetics of desmosomal protein accumulation at the junctions are affected by the absence of the desmoplakin-interacting protein ninein (Lechler & Fuchs, 2007) or by the ninein-dependent alteration of the microtubule network. However, with time, desmosomal structures are nevertheless formed because a large number of desmosomal proteins are still seen accumulated at the cellular cortex, including desmoplakin, desmoglein 3, and plakoglobin. But these desmosomal structures in the epidermis of *ninein* knockout mice are reduced in size and in density and display ultrastructural defects. Another ninein-dependent abnormality that might enhance desmosomal defects in keratinocytes manifests in slightly irregular positioning of nuclei in keratinocyte cultures (data not shown). Nuclear positioning has been demonstrated to depend on microtubules and ninein-related proteins (Stewart et al, 2015; Wang et al, 2015; Kowanda et al, 2016), and experimental alterations of nuclear positioning affect the assembly of desmosomes and weaken cellular adhesion between keratinocytes (Stewart et al, 2015). Altogether, the finding that desmosomal proteins, together with ninein, participate in the organization of cortical microtubule arrays (Lechler & Fuchs, 2007) hints at the existence of a positive feedback loop, in which cortical microtubules can in turn enhance the accumulation of desmosomal protein complexes, by providing tracks for local transport. Moreover, the substantial loss of desmoglein 1 from the *ninein*-KO epidermis may also contribute to desmosomal defects. Interestingly, during stratification of the skin, expression of different subtypes of desmosomal cadherins is tightly controlled and restricted to a particular skin layer: desmoglein 1 is expressed in the upper layers of the epidermis, whereas desmoglein 3 is expressed in the cells of the basal and immediate suprabasal layers (Mahoney et al, 1999). Because desmosomal cadherins play nonredundant, specific roles in tissue homeostasis, the reduced levels of desmoglein 1 in the ninein-knockout skin may affect desmosome assembly or stability specifically in the suprabasal layers, as seen in our experiments. Loss of desmoglein 1 may be due to reduced protein stability in the absence of ninein because the gene expression of desmoglein 1 is unaffected (Fig S2D).

## Reduced levels of late differentiation markers and changes in cellular morphology in the epidermis of *ninein*-knockout mice

In comparison with control mice, the epidermis from *ninein*-KO mice contains lower protein amounts of the differentiation markers filaggrin and involucrin. On the one hand, this may be explained by an altered timing of stratification and differentiation, after the aforementioned abnormalities in oriented cell division. On the other hand, these changes may be a consequence of altered epidermal homeostasis, after ninein-dependent defects in the microtubule cytoskeleton and in intercellular contacts. Evidence for the involvement of microtubules was provided in a recent study in which treatment of reconstructed human epidermis with Th2 cytokines, as identified in inflammatory skin disorders, altered epidermal homeostasis and lowered the amount of filaggrin (Hsu et al, 2018). After microtubule stabilization, these defects were reversed. In addition, cytoskeletal changes in ninein-deficient epidermis may lead to altered keratinocyte morphogenesis, and the latter may cause feedback on keratinocyte differentiation markers. For example, it has been shown that keratinocyte cell shape may control keratinocyte differentiation, whereby cells with a smaller contact surface increase the protein amounts of involucrin (Watt et al, 1988). Microtubules are directly involved in keratinocyte morphogenesis because microtubule-severing by overexpression of spastin in suprabasal keratinocytes leads to cell rounding and prevents the initial flattening of keratinocytes in the epidermis (Muroyama & Lechler, 2017). Consistent with Watt et al (1988), this is accompanied by increased expression of differentiation markers (Muroyama & Lechler, 2017). In our mice, the absence of ninein rather provokes the opposite effect, with increased flattening of keratinocytes leading to a reduced epidermal thickness and correlating with reduced levels of differentiation markers. Although we have no evidence for altered gene expression in the absence of ninein, protein stability of these markers may be altered (Fig S2D). A similar link between ninein, microtubules, and epithelial morphogenesis has been observed earlier, in polarized cells from simple epithelia, where ninein is needed to organize cortical apicobasal microtubule arrays and to enable cell elongation (Goldspink et al, 2017).

## Delayed formation of the skin barrier in *ninein*-knockout mice

Altogether, the possible delay in epidermal differentiation, combined with reduced filaggrin-dependent moisturizing activity, with defects in desmosome assembly and with defects in lamellar body secretion may explain why the formation of the skin barrier is delayed during the embryonal development in *ninein*-knockout mice. The formation of the skin barrier starts in a patterned manner, initiated at the dorsal region of mouse embryos (Hardman et al, 1998). Although the outside-in barrier of dorsal skin is almost complete at day E17.5 in wild-type embryos, most of the barrier is still incomplete in *ninein*-knockout mice at this time. Although *ninein*-knockout embryos seem to catch up with the completion of the barrier by day E18.5, minor defects remain because increased TEWL ("inside-out") was observed in newborn mice. In this context, the presence of loricrin may limit ninein-dependent defects: loricrin represents the bulk of CE protein and remains unaffected by

*ninein*-knockout (Fig S2C). It possesses barrier-promoting activities and may be compensating partially for the reduced expression of filaggrin and involucrin.

## Functional redundancy of other proteins with ninein

Epidermal phenotypes resulting from *ninein*-knockout and from tissue-specific *Lis1*-knockout show similarities, such as loss of cortical microtubule attachment, loss of desmoglein 1, reduced size and numbers of desmosomes, defects in the CE, and barrier defects (Sumigray et al, 2011). Nevertheless, the epidermal defects in ninein-knockout mice appeared much weaker as compared with the *Lis1*-knockout that resulted in perinatal lethality. *Ninein*-knockout mice, both with tissue-specific or constitutive knockout, were viable, and the barrier defects were relatively mild. This might be due to a broader range of functions that depend on Lis1, in addition to microtubule organization. Moreover, it is possible that a residual number of microtubules are still able to contact the cortex and that the loss of ninein is compensated by other proteins with partially redundant function. In vertebrates, ninein-like protein (Nlp) shares homology with ninein, in particular in its amino-terminal domain (Casenghi et al, 2003). Because mRNA levels of *Nlp* were unaffected by the knockout of *ninein*, we do not think that Nlp plays a major compensatory role (Fig S2D). However, detailed functional studies on Nlp have so far been limited because we were unable to find specific antibodies that recognize Nlp in mouse epidermis. In other experimental systems, such as *Drosophila* or *C. elegans*, ninein-related proteins were identified that were non-essential for viability in adults (Wang et al, 2015; Kowanda et al, 2016; Zheng et al, 2016). Interestingly, the *C. elegans* homologue NOCA-1 was shown to organize larval epidermal microtubule arrays, redundantly with the microtubule minus-end–binding protein patronin (Wang et al, 2015). Double loss-of-function mutants of *NOCA-1* with *patronin* resulted in defective apical transport of proteins, required for cuticle and barrier formation of *C. elegans* (Wang et al, 2015). In cells from simple epithelia of mice, the patronin-related proteins CAMSAP2 and 3 bind to the sites of microtubule organization at the apical cortex, even if ninein is delocalized from these sites, again suggesting functional redundancy (Noordstra et al, 2016; Toya et al, 2016; Goldspink et al, 2017).

Differing from our *ninein*-knockout phenotype, transgenic mice that overexpressed the microtubule-severing protein spastin in suprabasal layers contained keratinocytes with a disrupted microtubule network but kept an intact skin barrier (Muroyama & Lechler, 2017). This may be explained by a non–cell-autonomous effect because spastin was overexpressed only in part of the epidermal cells, and non–spastin-overexpressing keratinocytes might have compensated for epidermal defects by hyperproliferation and increased expression of filaggrin.

In summary, we provide evidence for two roles of ninein in epidermal morphogenesis, in the regulation of spindle orientation and in the organization of cortical microtubules. We are currently determining whether these cellular functions of ninein are of importance for any other tissues during development and whether they are of significance for the pathogenesis of Seckel syndrome. The mild barrier defects in the skin of newborn *ninein*-knockout mice are reminiscent of defects found in skin disorders such as

atopic dermatitis, where reduced protein levels of filaggrin and defects in the cornified layer have been implicated in pathogenesis (Cabanillas & Novak, 2016). A different set of more severe human skin disorders, but with molecular defects that resemble our *ninein*-knockout, have been reported from patients with mutations in the genes encoding desmoglein 1 or desmoplakin, leading to desmosomal defects and to dermatitis (Samuelov et al, 2013; Has et al, 2015; McAleer et al, 2015; Cheng et al, 2016; Schlipf et al, 2016). It will, thus, be of great importance for biomedical research to uncover the molecular mechanisms of epidermal microtubule organization, to determine the involvement of ninein in tissue homeostasis, and to identify any compensating mechanisms that contribute to the maintenance of the skin barrier.

# Materials and Methods

### Generation of ninein-deficient mice

Animals were housed and handled in accredited facilities, following the guidelines of the institution agreement no. #D315551 and project agreement no APAFIS#2725-20l51 1 1213203624 v5. The *ninein*-cKO and KO lines were established at the Genotoul Anexplo facilities of the "Centre de Biologie du Développement," University Toulouse III. Three confirmed positive ES-cell clones for the homologous recombination event of the *NinTM1A* allele were purchased from the Helmholtz Zentrum and injected into the C57Bl6 blastocysts. Several chimeras were obtained; one transmitted the *ninein TM1A* transgene through the germline, generating mouse founders (WT/TM1A). Genomic DNA from these mice was analyzed by PCR and by Southern blot, using internal probes (*lacZ* and *neomycin*) to confirm the presence of the target gene and the absence of deletions or duplications in the flanking 5′ and 3′ region. Transgenic mice were afterwards crossed with mice expressing ubiquitous FLP-recombinase (ID: 4673 IR2215-3/Rosa26-Deleter-Flp mice purchased from CERBM GIE, ICS, IGBMC), to induce recombination between both FRT-sites and the excision of the *lacZ* and the *neomycin* genes, which were located between the FRT-sites. The resulting offspring mice had exon 2 of ninein flanked with loxP sites (Fl/WT). These mice were finally mated to mice carrying Cre-recombinase (*K14*-Cre and *PGK1*-Cre), to induce tissue-specific or constitutive recombination between both loxP sites and thus creating the deletion of *ninein* exon 2 (Del). *K14*-Cre mice and *PGK1*-Cre mice have been previously described (Lallemand et al, 1998; Indra et al, 2000).

### PCR analysis of ninein genotypes

Extracts of F1 and F2 animals were prepared from tails of 3-wk-old offspring and extracts of F3 animals were obtained from the tails, hind legs, liver, or from newborn pups, E16.5, E17.5, or E18.5 embryos. The PCR primers (Fig 1A–C) were from intron 1 upstream of exon 2 (A 5′-TTTCTGAACGTCAGCCTGGATCGC-3′), exon 2 (B 5′-GTTGTTCTCCTCCTTCATCTCTCC-3′) and (D 5′-GAGGAGTCAGGTAAGAGCACTACC-3′); *lacZ* selection gene (C 5′-CAACGGGTTCTTCTGTTAGTCC-3′), and from the loxP site downstream of exon 2 (E 5′-TGAACTGATGGCGAGCTCAGACC-3′). Primers for the amplification of sequences from *K14*-Cre were

5′-GGAAAGTGTAGCCCGCAGGCC-3′ and 5′-ACAATCAAGGGTCCCCAAAC-TCACCC-3′ and primers for *PGK1*-Cre were 5′-CCATCTGCCACCAGC-CAG-3′ and 5′-TCGCCATCTTCCAGCAGG-3′. All primers were purchased form IDT. The PCR mix consisted of 1x coral buffer (Taq PCR Core kit; QIAGEN), 10 mM dNTP, 0.05% DMSO, 1 U Taq polymerase, 0.25 mM of each primer (IDT), and 10 ng genomic DNA. The PCR cycling conditions were 95°C for 3 min; followed by 35 cycles of 95°C for 30 s, 62°C for 30 s, and 72°C for 30 s (Mastercycler gradient, Eppendorf). PCR products were resolved on 2% agarose gels.

### Cell culture and treatment

The epidermal keratinocyte progenitors from mouse (MPEK) were purchased from CELLnTEC, and were grown in DermaLife complete medium at 35°C and 5% $CO_2$. Primary keratinocytes were obtained from the skin dissected from newborns. The skin was obtained from newborns and placed in DermaLife medium + dispase (2.4 U/ml) overnight at 4°C to separate epidermis and dermis. The epidermis was then incubated in 0.3% trypsin at 4°C for 15 min. The material was passed through a 40-$\mu$m cell strainer to obtain individual, dissociated cells and put in culture in DermaLife complete medium. To induce differentiation of the MPEK cell line, or of primary keratinocytes, 1.2 mM $CaCl_2$ was added to the medium 24 h before fixation.

MEFs were obtained from E14.5 embryos and cultured in DMEM supplemented with 10% FCS.

In the MPEK cell line, ninein was depleted using double-strand siRNA with the target sequences 5′-GCAUUAAUACUUAUCUUGUtt-3′ (Ambion s70605) and 5′-GCAUUCUAAGCUACAAUGAtt-3′ (Ambion s70606; data not shown for the latter). siRNAs were transfected using Lipofectamine RNAi max (Invitrogen). Two rounds of transfections every 48 h were required for proper depletion of the protein.

For disruption of the cytoskeleton, nocodazole (10 $\mu$M) was added 24 h after induction of differentiation for 5 h.

HeLa cells were cultured in DME supplemented with 10% FBS and grown at 37°C and 5% $CO_2$. Ninein was depleted using ds RNA oligomers 5′-UAUGAGCAUUGAGGCAGAG-3′ according to Dammermann and Merdes, 2002.

To test for any dominant effects of ninein amino acids 1–61, a mega-oligomer encoding the corresponding mouse sequence was cloned into pEGFP-C2. Mouse C2C12 cells grown in DMEM + 20% fetal calf serum were transfected using Fugene HD (Promega), according to the manufacturer's protocol.

### Flow cytometry

Primary cells were isolated incubating mouse embryonic skins on dispase (0.5 ml drops) (CELLnTEC) for 1 h at room temperature. After removal of the dermis, the epidermis was incubated under agitation in cold 0.25% trypsin (2 ml) for 5 min at 4°C and filtered through 100-micron cell strainers. Trypsin action was stopped by adding 3 ml DMEM plus 10% FCS. The cells were centrifuged for 5 min at 259 *g* at 4°C. The cells were resuspended in a mixture of ice-cold PBS and 70% ethanol and placed on ice for 2 h. The cells were then centrifuged for 5 min at 180 *g* and the ethanol solution was removed. After an additional wash in PBS, the cell pellet was suspended in a 1-ml solution containing 0.1% Triton X-100 (EMD

Millipore) in PBS, 2 mg DNase-free RNase A (QIAGEN) and 200 ml of 1 mg/ml propidium iodide (Sigma-Aldrich) and incubated at room temperature for 30 min. Acquisition was then performed on these samples using a FACSCalibur analyzer (BD) with a minimum of $1 \times 10^4$ cells analyzed with CellQuest Pro software (BD).

### EdU incorporation

Pregnant mice (16 d after plug) were injected with 25 $\mu$g EdU/g of animal weight, for 1 h (Fisher). E16.5 embryos were then dissected and their skin embedded in OCT and further processed for cryosections. Immunofluorescence was performed with antibodies against K10. EdU was visualized using "Click-iT kit" (Fisher).

### Tissue embedding and sectioning

Back-skin pieces or shoulder were obtained from embryos or newborns and processed for embedding with OCT or paraffin. Tissue that was soaked in OCT at room temperature was subsequently frozen, and 5-$\mu$m sections were cut with a cryostat. Alternatively, the tissue was fixed in 4% PFA before dehydration and paraffin embedding. Blocks were sectioned at 5 $\mu$m and the sections were rehydrated before use.

### Immunofluorescence microscopy

MPEK cells, primary keratinocytes, or MEFs were grown on coverslips and fixed in methanol at −20°C for at least 5 min and processed for immunofluorescence. Briefly, the cells were blocked in PBS, containing 0.1% Triton X-100 and 1% BSA or 0.5% fish gelatin, and incubated with antibodies in the same buffer. Paraffin or OCT sections were processed similarly. Before immunofluorescence, the paraffin sections were unmasked by incubation in 1% citrate or 50 mM glycine, pH 3.5, at 90°C for 40 min, rinsed in fresh PBS, and processed for immunostaining. Cryosections were post-fixed in methanol at −20°C before immunostaining. For Lis1-staining, fixation was performed in acetone at −20°C, for at least 5 min. For whole-mount immunofluorescence, the embryos (E13.5) were dissected and processed as previously described (Ezratty et al, 2011).

Images were acquired on widefield microscopes: Axiovert 200 m, Carl Zeiss, equipped with a Z motor, using a 63 × 1.4 NA objective and an AxioCam MRm camera with AxioVision software, or DeltaVision RT, using 40 × 1.3 NA, or 60 × 1.42 NA objectives, a CoolSnap camera, and SoftWoRx software for deconvolution, following the "ratio conservative" protocol. Image stacks for Figs S3A and S4B were acquired with a Leica TCS SP8 confocal microscope DMI6000, using a 40 × 1.3 NA objective on "resonant" mode and using LAS X software. Images for Figs 2A, 6C, and S4A were acquired using a Nikon Eclipse 80i widefield microscope using 20 × 0.5 NA or 40 × 1.3 NA objectives, and a Nikon DXM1200 camera, with NIS Elements AR software.

### Image processing and quantifications

For image processing and quantification of fluorescence intensities, ImageJ software was used. Intensities were measured in image stacks acquired with constant exposure settings, after maximal projection. For measurements of centrosomal protein intensity, a

circular area (1.5 $\mu$m diameter) around centrosomes was used, based on pericentrin staining. For background correction, the mean intensity of an adjacent area in the cytoplasm was subtracted. To determine the distribution of microtubules or Lis1 around the cell cortex, a 10 × 3 $\mu$m rectangle was drawn orthogonally to the line of $\alpha$- or $\beta$-catenin staining. The intensity profile was obtained using the plot profile tool of ImageJ. Multiple cell measurements were aligned, using the maximum intensity of the catenin channel as a reference for the cell membrane, and averaged. Desmosome length and the area of lamellar body secretion were measured with the ImageJ measurement tool. Desmosome numbers and Ki67-positive cells were counted manually on regions acquired by electron microscopy or immunofluorescence microscopy. Angle measurements of spindles in cultured cells were performed after acquisition of z stacks, covering the total volume of the spindle, followed by rotation around the x axis with the "3D project" tool (Fiji, ImageJ). A line was drawn, connecting the two poles, as identified by centrin staining, and the angle was measured between this line and the coverslip. Angle measurements of spindles in skin were performed on E16.5 OCT thick sections (30 $\mu$m), after staining with antibodies against NuMA and H3P. Stacks were rotated in Imaris, to obtain the spindle axis in the x–y plane. Angles were measured in ImageJ, between a line connecting the two poles (identified by NuMA staining) and the interface between dermis and epidermis. Primary cilia were counted manually in basal or suprabasal z planes from confocal stacks, obtained from whole-mount epidermis, stained for E-cadherin, ODF2, and acetylated tubulin.

## Western blotting

For Western blotting on MEFs, the cells were washed in PBS and lysed (50 mM Tris–HCl, pH 7.5, 150 mM NaCl, 2 mM EDTA, 0.5% Triton-X100, 1 mM DTT, and protease inhibitor cocktail) for 10 min on ice. Clear lysates were obtained by centrifugation for 10 min at 16,000 $g$ at 4°C. For Western blotting of epidermal proteins, frozen skin samples were homogenized in extraction buffer (40 mM Tris–HCl, pH 7.4, 10 mM EDTA, 50 mM DTT, 8 M urea, 0.25 mM PMSF, and protease inhibitor cocktail), using a Potter homogenizer on ice. Clear lysates were obtained by centrifugation for 15 min at 15,000 $g$ at 4°C. The samples were prepared for SDS–PAGE, by boiling in sample buffer. Equal amounts of protein were loaded. Proteins were transferred onto nitrocellulose membranes and probed with antibodies. After incubation with primary antibodies and rinsing, secondary antibodies conjugated to IRDye 800CW and 680CW (Invitrogen) were added. An Odyssey Imaging System (LI-COR Biosciences) was used to detect protein signals.

## CE isolation and Nile red staining

Pieces of newborn mouse dorsal epidermis were boiled at 95°C with agitation for 10 min in cornified envelope dissociation buffer (100 mM Tris–HCl, pH 8.5, 5 mM EDTA, 20 mM DTT, and 2% SDS). CEs were pelleted by centrifugation for 10 min at 12,000 $g$. Extraction with dissociation buffer was repeated three times, and finally CEs were resuspended in 100 mM Tris–HCl (pH 8.5), 5 mM EDTA, 20 mM DTT, and 0.2% SDS. CE suspensions were dropped onto microscopy slides and air-dried. After 10 min acetone fixation, and following rehydration with PBS, the CEs were stained with a Nile red solution [(9-(diethylamino)benzo[a]phenoxazin-5(5H)-one), Sigma-Aldrich] at 0.5 $\mu$g/ml in 75% glycerol.

## Transmission electron microscopy

Mouse skin was fixed with 2.5% glutaraldehyde and 2% para-formaldehyde in 0.1 M cacodylate buffer, pH 7.2 (EMS), for 24 h at 4°C and post-fixed at 4°C with 1% $OsO_4$ and 1.5% $K_3Fe(CN)_6$ in the same buffer. The Samples were treated for 1 h with 1% aqueous uranyl acetate, dehydrated in a graded series of acetone, and embedded in EMbed 812 resin (EMS). After 48 h of polymerization at 60°C, the ultrathin sections (80 nm) were mounted on 75 mesh Formvar carbon–coated copper grids. The sections were stained with UranyLess (Delta Microscopies) and lead citrate. Grids were examined with a TEM (Jeol JEM-1400; JEOL Inc) at 80 kV. Images were acquired using a digital camera (Gatan Orius; Gatan Inc).

## Quantitative PCR

Total RNA was extracted from cell pellets (MPEK for depletion experiments, or MEFs) and skin samples from WT and KO mice, using an RNeasy Plus Mini kit (QIAGEN), according to the manufacturer's instructions. Equal amounts of RNA (500 ng) were added to a reverse transcriptase reaction mix (qScript cDNA Synthesis kit; Quanta Biosciences). The resulting cDNA was subjected to quantitative PCR using a Biorad C1000 thermal cycler, coupled to a CFX96 real-time system, using the SSoFast Evagreen Supermix (Bio-Rad) for 40 cycles. Calibration curves were performed on cDNA from each cell type, confirming Rps16 (ribosomal protein s16) and Tbp (TATA box–binding protein) as suitable reference genes, to measure the efficiency of the primer pairs used. Corresponding reaction mix containing RNA without reverse transcriptase was used as a negative control for the qPCR of each sample. Levels of individual cDNAs were displayed relative to those of Rps16 or Tbp, using the Bio-Rad CFX Manager software. Relative expression levels of mRNA were calculated using methods 2-ΔΔCts, or Pfaffl. The following primers were used: ninein forward: TGCTGCAACAGACGCTACTC; ninein reverse: TCAAAGTGCTCCTCACTGGA; Rps16 forward: AGGAGC-GATTTGCTGGTGTGG; Rps16 reverse: GCTACCAGGGCCTTTGAGATGG; ninein-like protein forward: GACCCCAGGTACCCATAGAAG; ninein-like protein reverse: GGCGTAGACTCACCCACAA. Primers for Tbp, filaggrin, desmoglein 1, and corneodesmosin were used as in Leclerc et al (2014).

## Epidermal barrier assay

Embryos were dissected in PBS and transferred to 4 ml staining solution (PBS, containing 5 mM $K_4[Fe(CN)_6]$; 6.4 mM $K_3[Fe(CN)_6]$; 2 mM $MgCl_2$; 0.01% sodium deoxycholate; 0.2% NP-40; and 100 mg/ml X-Gal) at 37°C for 12 h (Hardman et al, 1998). After staining, the embryos were rinsed in PBS and photographed. A total of 15 cKO, 17 KO, and 39 control embryos were analyzed in nine independent experiments.

### TEWL

41 KO and 26 WT newborns were analyzed in six independent experiments, using an Evaporimeter ("Vapometer" Delfin), according to the manufacturer's guidelines.

### Antibodies

The following primary antibodies were used: rabbit anti-NuMA EP3976, rabbit-anti-ODF2 ab43840, and rabbit anti-NOTCH3 ab23426 (Abcam); mouse anti-$\beta$-catenin 610153, mouse anti-plakoglobin 610253, mouse anti-p150-glued 612708, and mouse anti-pericentrin 611815 (BD Bioscience); mouse anti H3P (Cell Signaling Technology); rabbit anti-filaggrin PRB-417p, rabbit anti-involucrin PRB-140c, rabbit anti-loricrin PRB-145p, and rabbit anti-K5 PRB-160p (Covance Research Products); mouse anti-$\gamma$-tubulin TU-30 (Exbio); mouse anti-claudin 1 37-4900 (Invitrogen); mouse anti-desmoglein 3 D218-3 (MBL); mouse anti-actin MAB1501 (Merck Millipore); rabbit anti-Lis1 sc-15319, rabbit anti-desmoglein 1 sc-20114, and mouse anti-GAPDH sc-32233 (Santa Cruz Biotechnology); rabbit anti $\alpha$-catenin DECMA1, mouse anti-centrin 20H5, mouse anti-$\alpha$-tubulin T5168, mouse anti-acetylated tubulin T7451, mouse anti-K10 SAB4501656, mouse anti-K1-10/anti-pan-cytokeratin c2562, and mouse anti-desmoplakin 11-5F (Sigma-Aldrich); rabbit anti-Ki67 RM-9106 (Thermo Fisher Scientific); rabbit anti-ninein L77 (Delgehyr et al, 2005); rabbit anti-GCP2 (Haren et al, 2006); rabbit anti-$\gamma$-tubulin R75 (Julian et al, 1993); rabbit anti-ninein (Ou & Rattner, 2000); and mouse anti-corneodesmosin F27.28 (Serre et al, 1991).

### Statistical analysis and replication of experiments

Statistical analysis was performed using Prism 5 software. Two-tailed, unpaired $t$ tests were performed to compare experimental groups. The results are reported in the figures and figure legends. The immunofluorescence and Western blot images are representative of multiple experiments that were repeated with similar results. Quantitative PCR experiments were run in triplicates, from three independent experiments.

## Supplementary Information

## Acknowledgements

We thank Drs. P Chambon (GIE-CERBM, IGBMC, Illkirch) and V Bergoglio (IPBS, Toulouse) for the K14-Cre mice and D Morello (Centre de Biologie du Développement Toulouse) for the PGK-Cre mice. We thank the Genotoul service facilities, ABC Zootechnics, and the histopathology facilities, Toulouse Réseau Imagerie (TRI) light microscopy and electron microscopy facilities (S Balor and V Soldan) for their excellent support. We thank Drs. M Simon, G Serre, N Jonca, D Garrod, J Sillibourne, JB Rattner, and A Davy for antibodies. We thank A Davy, C Audouard, M Djabali, and P Mercier for their help with the generation of mutant mice, and T Jungas and T Mangeat for further technical help. This study was supported by research grants from the "Centre d'Etudes et de Recherche sur la Peau et les Epithéliums de Revêtements," Toulouse, and from the "Bonus Qualité Recherche" programme, University Toulouse III. N Lecland was supported by a long-term fellowship from the European Molecular Biology Organization (ALTF 1285-2014).

### Author Contributions

N Lecland: conceptualization, investigation, methodology, and writing—original draft, review, and editing.
C-Y Hsu: conceptualization, investigation, methodology, and writing—original draft, review and editing.
C Chemin: investigation, methodology, and writing—review and editing.
A Merdes: conceptualization, supervision, funding acquisition, investigation, and writing—original draft, review, and editing.
C Bierkamp: conceptualization, supervision, funding acquisition, investigation, and writing—original draft, review, and editing.

### Conflict of Interest Statement

The authors declare that they have no conflict of interest.

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
