## [Reviewer comments · Life Science Alliance]

Life Science Alliance

Epidermal development requires ninein for spindle orientation and cortical microtubule organization

Nicolas Lecland, Chiung-Yueh Hsu, Cécile Chemin, Andreas Merdes, and Christiane Bierkamp

DOI: <https://doi.org/10.26508/lsa.201900373>

Corresponding author(s): Andreas Merdes, University Paul Sabatier/CNRS and Christiane Bierkamp, University Toulouse

Review Timeline:

Submission Date:	2019-03-11
Editorial Decision:	2019-03-11
Revision Received:	2019-03-12
Accepted:	2019-03-12

Scientific Editor: Andrea Leibfried

Transaction Report:

Please note that the manuscript was previously reviewed at another journal and the reports were taken into account in inviting a revision for publication at *Life Science Alliance* prior to submission to *Life Science Alliance*.

Reviewer #1 Review

Comments:

In this manuscript Lecland and colleagues explore the importance of the centrosomal protein Ninein to cytoskeleton remodeling and epithelial differentiation during embryonic skin development. Using siRNA to perturb Ninein in mouse keratinocytes, the authors observe that upon differentiation by calcium induction, keratinocytes retain cytoskeletal undifferentiated properties. Microtubule (MT) reorganization is impaired and a defective recruitment of Lis1 to the junctions leads to the abnormal assembly of desmosomes. Furthermore, they observe that nuclei will fail their normal position within the differentiated cells. To further characterize Ninein function under physiological conditions, the authors use constitutive and conditional Ninein knock-out mice and observe that although viable, embryos and newborn pups have a thinner skin and an epidermal barrier defect that culminates with a significant water loss. They observe that desmosomes don't properly form and secretion of granular components is severely impaired. On the basis of these findings Lecland et al, propose a model where Ninein regulates epidermal differentiation through a mechanism that may not exclusively depend on its role in microtubule organization.

While Lecland discoveries nicely show the importance of Ninein to epidermal barrier formation, their results fail to rigorously show the mechanism by which that impairment occurs. Furthermore, their novelty on the role of Ninein to the granular layer formation, is overshadowed by the initial in vitro studies that are not adding much novelty to what has been previously described for Lis1 and Nde1/Ndel1. On that sense, I would recommend a major revision that solidify authors findings and novelty, before the paper can be accepted.

Major Points:

It has previously been shown that upon keratinocyte differentiation the cytoskeletal MT network will reorganize. Instead of radial distribution from centrosomes, MTs are cortically positioned near cell-cell junctions independent of the centrosome. Because of its MT anchoring function, Ninein assumes a fundamental role as it is recruited by desmoplakin to desmosomes. Lis1 and Nde1, two other centrosomal proteins, were also shown to be able to interact with Ninein and their perturbation impairs barrier function in mice embryos and newborn pups. It is thus not surprising that the authors would observe similar phenotypes to those that were previously reported. On that sense, the authors do not provide significant advance to the field with their in vitro observations and they do not show their physiological relevance. For instance, what is the consequence of mispositioned nuclei to the differentiated cells? The authors should consider to restructure the manuscript and use the KO keratinocytes to the in vitro characterization instead of the data with only one siRNA. Finally, I would strongly recommend that the authors challenge the system in the undifferentiated situation. Are cells able to properly migrate in a scratch wound assay? would the nuclear position be also affected? Would migration be perturbed? What about cell division? can cells divide properly? Is the spindle position stable after Ninein depletion? Because the authors have KO keratinocytes available they should be able to investigate these issues further.

The authors use 2 in vivo models, the constitutive and conditional KO and they nicely show that there is a reduction in the thickness of the layers. However they show that there is an increased proliferation of the suprabasal layer. While this observation seems counterintuitive the authors fail to investigate this issue properly. Quantification of the proliferation in the K5 and K10 layer should be accurately done: the authors should consider doing Edu injections and co-stain with K5 and K10

to accurately determine which cells are proliferating. And why is there an increase in proliferation? Are those cells stuck in mitosis? Also is the absence of Ninein perturbing symmetric/asymmetric cell division ratios? Are the axis of division randomized? Are the cells stuck in mitosis due to a putative MT/centrosome anchorage problem? What happens to those proliferating cells? Are they going through apoptosis or other kind of cell death and cleared out? This would be interesting to show. Importantly, the authors claim that the K10 layer is not perturbed and it is clear that the antibody did not work (Supplemental Figure 4).

The major breakthrough of this study is the observation that the granular layer is not properly assembled upon Ninein depletion. The authors should strengthen this observation and explore whether corneodesmosin secretion depends or not on the microtubules network. It would also be interesting to explore in the context of other centrosomal or MT associated proteins, whether similar secretion perturbations are observed. Furthermore, it still remains to be unveiled whether the lamellar bodies are not properly assembled due to a perturbation in the MT network of the granular layer, or whether the granular cells are not properly differentiated due to a previous perturbation in the K5/14 or K1/K10 layers. Unveiling the specific contribution of Ninein to the biology of the differentiation program will be a major finding and advance in the field.

The imaging quality and presentation can and should be improved. When the signal-to-noise ratio is low, the authors should consider inverting the color or increasing the contrast so the reader can better observe what the authors are attempting to describe (Lis1 staining Figure 2). Furthermore centrosome staining should be shown at higher magnification (Figure 1, Figure 3). More importantly, the authors should try to have images from their tissue sections aligned, with the basement membrane parallel to the longer axis of the figure rectangle.

Minor Points:

Figure 1E and Figure 2B. The graphs here presented are not easy to interpret. Less x points should be shown.

Figure 4E. The authors claim a dotted profilagrin staining that cannot be observed from the images provided

Figure 4E and 4F, Figure 5 and Figure 6 and Figure 7. Data for the Ninein cKO is missing and should be shown in parallel with the KO data (or at least provided in supplemental).

Figure 6A and 6B better images need to be provided with proper quantifications. The skin seems to have similar thickness.

Reviewer #2 Review

Comments to the Authors (Required):

Comments for the authors:

In this manuscript, Lecland et al explore the role of the conserved centrosomal protein Ninein in epidermal cell differentiation using in vitro and in vivo loss of function approaches. Overall, the authors document that loss of Ninein results in alterations in the cortical organization of MT, desmosomal assembly and nuclear positioning during keratinocyte differentiation. In vivo, the loss of Ninein leads to a histologically thinner epidermis, reductions in desmosome numbers, associated with slight alterations in epidermal barrier formation.

Although the authors document some interesting changes in epidermal differentiation upon loss of Ninein; I found the documented findings preliminary, since this manuscript lacks sufficient depth and mechanistic insights to provide a conceptual advance of the molecular mechanisms/signals mediated by Ninein in the control of epidermal differentiation.

Major comments:

1. Ninein in MT cortical organization and nuclear positioning. The authors describe that the effects of Ninein in these events are dependent on MTs, using MT depolymerizing agents. How is this regulated? The direct involvement of these connections should be validated using rescue experiments expressing mutant proteins.
2. Nuclear positioning and differentiation. The roles of Ninein, nuclear positioning and differentiation are not clearly connected with the effects on epidermal differentiation. Is this also observed in vivo?
3. Epidermal loss of Ninein in the epidermis, proliferation and differentiation balance. The authors document higher proliferation and less differentiation. Still, the phenotype observed is a thinner epidermis. How Ninein leads to an increase in proliferating cells? Moreover, since Ki67 levels oscillate with higher expression in G2, are these cells rather arrested and excluded to suprabasal layers. The authors should evaluate the mitotic index and the proliferation index in their samples, in particular during epidermal barrier development.
4. The differentiation of the spinous layer is not affected by the loss of Ninein expression in vivo. But rather the secretion of lamellar bodies, the reduction of filaggrin expression levels and corneodesmosin, What is the molecular mechanism regulated by Ninein underlying this effect?

Minor comments:

Fig 1C. The histograms have different Y bar distribution.

Fig2. The Lis1 immunostaining should be improved.

Differentiation markers.

Fig S4. The staining of K1 and K10 is not clear in both controls and samples.

Reviewer #3 Review

Comments to the Authors (Required):

In the manuscript "Epidermal differentiation and barrier defects in mice lacking microtubule-organizing protein Ninein," Lecland et al explore a role for Ninein in desmosomal integrity and how it can contribute to skin disorders such as dermatitis. Using a combination of RNAi and the generation of knockout mice, the authors' intent is to establish a link between the microtubule nucleation role of Ninein (evidenced by reduced Lis1 recruitment, microtubule anchoring at desmosomal sites, and reduced nuclear tethering) with differentiation defects in the epidermis (through hyperproliferative suprabasal layers, delayed granular layer formation, and impaired barrier). To link the two events, the authors provide data showing that the loss of Ninein impairs lamellar body secretion of corneodesmosin, resulting in defects in desmosome ultrastructure. On the surface, this study would be of interest to the Journal of Cell Biology, but the work is mostly descriptive and falls short of its goal in terms of caveats and execution.

Of greatest concern:

1. In studying the function of Ninein, a subdistal appendage protein, the authors primarily focus on the role of Ninein as microtubule anchoring protein. However, the ablation of the Ninein can

potentially affect the biogenesis and function of cilia, which require the appendages of centrosomes for docking at the plasma membrane. Importantly, the loss of cilia has been reported to inhibit epidermal differentiation during development, specifically generating defects in the filaggrin-positive granular layer, compromising the epidermal barrier, and resulting in aberrant proliferative cells in the suprabasal layer (Ezratty et al 2011). Consequently, the phenotypic defects described within the manuscript, encompassing almost all of Figure 4 and 7, could be a pleiotropic effect of both impaired ciliary function and microtubule anchoring defects at the desmosomes, and must be addressed. The authors need to carefully examine ciliary biogenesis, structure, and signaling in their Ninein KO/cKO animals. As suggestion, the authors need to quantify the number of cilia in wt versus Ninein KO, cKO tissue. They need to look at the ultrastructure of the cilia (i.e ciliary length by immunofluorescence, EM sections, etc). In addition, the authors need to investigate whether Notch signaling, an output of ciliary signaling in the epidermis, is affected in Ninein KO/cKO tissue. Should ciliary signaling be the primary cause of the differentiation defects observed in the epidermis, it would negate the primary hypothesis of this study, that microtubule anchoring at desmosomes contributes to differentiation defects. The authors would need to design rescue experiments splitting the two roles of Ninein, and in parallel, devise new experiments demonstrating the consequence of desmosomal defects and lamellar body secretions, and significantly strengthen those arguments.

2. Along the same line of compounding effects, the authors make a point to discuss that the centrosomal function of Ninein are unaltered. As point of comparison, the authors refer to several mouse epithelial models of Plk4 overexpression, in which the observed phenotypes are different than the ones observed for the Ninein KO. This comparison is inaccurate, because Plk4 overexpression induces centrosome amplification rather than structural defects of the centrosome. Consequently, the phenotypes would be different as well. The more likely point of comparison for relating phenotypic changes would be mutations or ablations of centrosome appendage proteins, which would affect mother centriole activities. In that regards, defects in asymmetric cell divisions and ciliogenesis could further compound the phenotype and would again be an obvious culprit to investigate (see also point 1). Examination of asymmetric cell divisions should be more flushed out. Other major concerns:

3. The manuscript could be organized in a more concise manner and better packaged as a report. The primary novelty of the study is the report of the first Ninein knockout mice. From that standpoint, Figures 1 and 2, which utilize shRNA knockdown, do not significantly add to the studies. Could the authors not have conducted the microtubule organization studies using the knockout primary epidermal progenitor keratinocytes they established in culture, and provided the shRNA studies as support in supplemental figures? It is also unclear how the tethering of the nuclei (Figure 2C-E) contribute to desmosomal defects.

4. It is unclear whether the authors effectively achieve their desired Ninein knockout, because a more thorough characterization of the animals is missing from the figures. The authors report that the straight knockout animal (from PGK-Cre x Ninein fl/fl mating) are viable, with no further reporting of other phenotypic defects. This raises concern because two other groups have reported that Ninein is required for the development of neuronal lineages in vertebrate organisms (Dauber et al 2012, Wang et al 2009). If the animals are viable, are they born with Mendelian ratios? Do they have brain development defects? Do they display polydactyly? If they don't, is the entirety of the Ninein protein ablated in these animals? The authors do provide a western blot and IF images of the loss of Ninein in Figure 3D/G. However, they use a C-terminal antibody to recognize the protein, so it is unclear which isoforms this antibody recognizes, and whether there is a truncated Ninein isoform remaining which alleviates some of their results. Could the authors use a larger panel of qPCR primer sets to detect Ninein and its isoform variants, spanning several exons junctions and, importantly, Exon1?

5. The authors claim that the Ninein KO and cKO mice have flaky skin. The authors should provide

evidence of this claim, in the form of an image of the flaky skin on the pups.

6. The assays utilized to demonstrate defects in the differentiation of the epidermis (Figure 4) could be improved to be more informative. For example, the authors quantify the thickness of the epidermis based on H&E staining (Figure 4A/B), even though the stain itself looks different between the WT/Ninein-cKO and Ninein KO panels. It would be more informative if the authors quantified each compartment (basal, spinous, granular) based on compartment-specific markers (K5, K10, Involucrin). The K1/K10 staining of the epidermis looks like it is staining the granular layers rather than the spinous layers (Figure S4A) and calls into question the validity of the antibody. The K5 layer (Figure S4A) looks like there are two-cells thick in certain regions. Lastly, the authors should use EDU labeling counterstained with the K5 marker to understand the significance of the suprabasal proliferative cells in Figure 4C. If those cells are K5+, then they likely represent hyperproliferation of the basal layer, another attribute of ciliary dysfunction in the epidermis.

Of minor concern:

7. The text sometimes suffers from lack of clarity, grammatical errors, inaccurate diction, changing fonts.

8. The legend and description of Figure 3C within the text are much too confusing to figure out and could benefit from simplification.

9. The western blot in Figure 3G needs protein markers to delineate the size of the observed protein.

10. In figure S3C, it is unclear what the arrows are pointing to. In the same figure legend, there is mention of a dotted line, but only a solid line is present on the image.

11. In figure S3D, the staining is suboptimal, making the interpretation of the p150-glued data challenging.

Response to the reviewers:

Reviewer #1

“...what is the consequence of mispositioned nuclei to the differentiated cells? The authors should consider to restructure the manuscript and use the KO keratinocytes to the in vitro characterization instead of the data with only one siRNA. Finally, I would strongly recommend that the authors challenge the system in the undifferentiated situation. Are cells able to properly migrate in a scratch wound assay? would the nuclear position be also affected? Would migration be perturbed?”

(See also response to reviewer 2, point 2) In an attempt to investigate a potential role of ninein in nuclear positioning during epidermal development in vivo, we compared nuclear positions in mouse tissue. Due to the small ratio of cytoplasm versus nuclear volume in suprabasal cell layers, we were technically unable to obtain solid information on centroid versus non-centroid nuclear positions. Since our previous experiments were exclusively obtained in cultured cells, and since the mechanisms that link nuclear positioning to the formation of cell adhesions in keratinocyte cultures and hair follicles were already investigated in a previous publication by Stewart et al (2015, J Cell Biol 209:403-18), we decided to drop them from this manuscript, except for a short comment on page 19.

“What about cell division? can cells divide properly? Is the spindle position stable after Ninein depletion? Because the authors have KO keratinocytes available they should be able to investigate these issues further. The authors use 2 in vivo models, the constitutive and conditional KO and they nicely show that there is a reduction in the thickness of the layers. However they show that there is an increased proliferation of the suprabasal layer. While this observation seems counterintuitive the authors fail to investigate this issue properly. Quantification of the proliferation in the K5 and K10 layer should be accurately done: the authors should consider doing Edu injections and co-stain with K5 and K10 to accurately determine which cells are proliferating. And why is there an increase in proliferation? Are those cells stuck in mitosis? Also is the absence of Ninein perturbing symmetric/asymmetric cell division ratios? Are the axis of division randomized? Are the cells stuck in mitosis due to a putative MT/centrosome anchorage problem? What happens to those proliferating cells? Are they going through apoptosis or other kind of cell death and cleared out?”

We added an entire set of new data that demonstrate that spindle orientation is irregular in the absence of ninein, due to the loss of astral microtubules. We demonstrate spindle orientation defects both in cell cultures and in vivo, in the basal layer of the epidermis. We quantified the effects on the basal progenitor pool, on differentiation, on proliferation, mitotic index, and apoptosis. New data, also including EdU labeling experiments, have been added to Figure 2C, Figure 3, and supplementary Figures S3, and S4.

“Importantly, the authors claim that the K10 layer is not perturbed and it is clear that the antibody did not work (Supplemental Figure 4).”

Using new antibodies, we repeated this experiment and added the new data to Figure 2C.

“The major breakthrough of this study is the observation that the granular layer is not properly assembled upon Ninein depletion. The authors should strengthen this observation and explore whether corneodesmosin secretion depends or not on the microtubules network. Furthermore, it still remains to be unveiled whether the lamellar bodies are not properly assembled due to a perturbation in the MT network of the granular layer, or whether the granular cells are

not properly differentiated due to a previous perturbation in the K5/14 or K1/K10 layers.”

We have added new data to Figure 6 and S2D, showing the effect of ninein-KO on corneodesmosin and lamellar body secretion, whereas earlier steps of corneodesmosin expression and lamellar body assembly (EM in Fig. 6 F+G) remain unaffected.

“Unveiling the specific contribution of Ninein to the biology of the differentiation program will be a major finding and advance in the field.”

We have added new data on Notch signaling and on ciliogenesis in suprabasal keratinocytes (Fig. S4), and data on the expression of various differentiation markers (Fig. S2D). Altogether, these data indicate that the differentiation program is largely normal in ninein-KO epidermis. Defects in differentiated cells are specifically seen at the level of protein abundance of selected markers, at the level of desmosome assembly, and lamellar body secretion.

“The imaging quality and presentation can and should be improved. When the signal-to-noise ratio is low, the authors should consider inverting the color or increasing the contrast so the reader can better observe what the authors are attempting to describe (Lis1 staining Figure 2).”

We replaced various figures with better images, and we inverted the contrast of Lis1 staining in the new Figure 4F.

“Furthermore centrosome staining should be shown at higher magnification (Figure1, Figure 3). More importantly, the authors should try to have images from their tissue sections aligned, with the basement membrane parallel to the longer axis of the figure rectangle.”

Besides substituting for various new images, we aligned nearly all tissue sections as suggested, with the exception of Fig. 1G (which is oriented slightly oblique, to reveal details on centrosomal staining in the dermis).

Minor Points:

“Figure 4E. The authors claim a dotted profilaggrin staining that cannot be observed from the images provided”

We have added new images (new Fig. 2D) and we have slightly modified the description in the corresponding text.

“Figure 4E and 4F, Figure 5 and Figure 6 and Figure 7. Data for the Ninein cKO is missing and should be shown in parallel with the KO data (or at least provided in supplemental). Figure 6A and 6B better images need to be provided with proper quantifications. The skin seems to have similar thickness.”

We have added data on ninein cKO to the new supplementary Figures S2 B, C, S5 D, E, and S6. Images of corneodesmosin were replaced by new ones (new Fig. 6A). Quantifications can be found in the corresponding text on page 13.

Reviewer #2:

Major comments:

“1. Ninein in MT cortical organization and nuclear positioning. The authors describe that the effects of Ninein in these events are dependent on MTs, using MT depolymerizing agents. How is this regulated? The direct involvement of these connections should be validated using rescue experiments expressing mutant proteins.”

Our revised manuscript provides now better controlled data on a role of ninein in non-centrosomal, cortical microtubule organization. This is the first manuscript that shows directly a role of ninein in the epidermis. We agree that investigating protein interactions of ninein with the help of mutant ninein would be a good approach to investigate these problems in more detail. Because of limited manpower, time, and budget, we were not able to complete such experiments for a revised manuscript, but we would be interested in exploring this in the future.

“2. Nuclear positioning and differentiation. The roles of Ninein, nuclear positioning and differentiation are not clearly connected with the effects on epidermal differentiation. Is this also observed in vivo?”

(See also response to reviewer 1, point 1) In an attempt to investigate a potential role of ninein in nuclear positioning during epidermal development in vivo, we compared nuclear positions in mouse tissue. Due to the small ratio of cytoplasm versus nuclear volume in suprabasal cell layers, we were technically unable to obtain solid information on centroid versus non-centroid nuclear positions. Since our previous experiments were exclusively obtained in cultured cells, and since the mechanisms that link nuclear positioning to the formation of cell adhesions in keratinocyte cultures and hair follicles were already investigated in a previous publication by Stewart et al (2015, J Cell Biol 209:403-18), we decided to drop them from this manuscript, except for a short comment on page 19.

“3. Epidermal loss of Ninein in the epidermis, proliferation and differentiation balance. The authors document higher proliferation and less differentiation. Still, the phenotype observed is a thinner epidermis.

How Ninein leads to an increase in proliferating cells? Moreover, since Ki67 levels oscillate with higher expression in G2, are these cells rather arrested and excluded to suprabasal layers. The authors should evaluate the mitotic index and the proliferation index in their samples, in particular during epidermal barrier development.”

(See also response to reviewer 1) We added an entire set of new data that demonstrate that spindle orientation is irregular in the absence of ninein, due to the loss of astral microtubules. We demonstrate spindle orientation defects both in cell cultures and in vivo, in the basal layer of the epidermis. We quantified the effects of ninein KO on the basal progenitor pool, on differentiation, on proliferation, mitotic index, and apoptosis. New data, also including EdU labeling experiments, have been added to Figure 2C, Figure 3, and supplementary Figures S3, and S4.

Moreover, we have added new data on Notch signaling and on ciliogenesis in suprabasal keratinocytes (Fig. S4), and data on the expression of various differentiation markers (Fig. S2D). Altogether, our data indicate that the amount of basal progenitor cells is reduced, and that the effects of ninein KO on suprabasal cell proliferation are minor. The differentiation program is largely normal in ninein-KO epidermis. Defects in differentiated cells are specifically seen at the level of protein abundance of selected markers, at the level of desmosome assembly, and lamellar body secretion.

“4. The differentiation of the spinous layer is not affected by the loss of Ninein expression in vivo. But rather the secretion of lamellar bodies, the reduction of filaggrin expression levels and corneodesmosin, What is the molecular mechanism regulated by Ninein underlying this effect?”

We propose a specific effect of the ninein-dependent cortical microtubule network, on the localized transport of cortical material, such as during desmosome assembly or secretory processes. We show now by qPCR that the expression of epidermal differentiation markers is largely normal (new supplementary Fig. S2D), and that differentiation-dependent processes such as lamellar body assembly (but not secretion) are normal (Fig. 6D-G).

Minor comments:

“Fig 1C. The histograms have different Y bar distribution.”

In the revised manuscript, we have largely replaced data based on siRNA by data based on ninein KO. For this reason, the mentioned graph on the silencing of ninein mRNA has been removed.

“Fig2. The Lis1 immunostaining should be improved.”

We have inverted the contrast, as suggested by reviewer 1 (new Fig. 4F).

“Fig S4. The staining of K1 and K10 is not clear in both controls and samples.”

We have substituted these images with new data, using new antibodies, now in Fig. 2C.

Reviewer #3:

“1. In studying the function of Ninein, a subdistal appendage protein, the authors primarily focus on the role of Ninein as microtubule anchoring protein. However, the ablation of the Ninein can potentially affect the biogenesis and function of cilia, which require the appendages of centrosomes for docking at the plasma membrane. Importantly, the loss of cilia has been reported to inhibit epidermal differentiation during development, specifically generating defects in the filaggrin-positive granular layer, compromising the epidermal barrier, and resulting in aberrant proliferative cells in the suprabasal layer (Ezratty et al 2011). Consequently, the phenotypic defects described within the manuscript, encompassing almost all of Figure 4 and 7, could be a pleiotropic effect of both impaired ciliary function and microtubule anchoring defects at the desmosomes, and must be addressed. The authors need to carefully examine ciliary biogenesis, structure, and signaling in their Ninein KO/cKO animals. As suggestion, the authors need to quantify the number of cilia in wt versus Ninein KO, cKO tissue. They need to look at the ultrastructure of the cilia (i.e ciliary length by immunofluorescence, EM sections, etc). In addition, the authors need to investigate whether Notch signaling, an output of ciliary signaling in the epidermis, is affected in Ninein KO/cKO tissue. Should ciliary signaling be the primary cause of the differentiation defects observed in the epidermis, it would negate the primary hypothesis of this study, that microtubule anchoring at desmosomes contributes to differentiation defects. The authors would need to design rescue experiments splitting the two roles of Ninein, and in parallel, devise new experiments demonstrating the consequence of desmosomal defects and lamellar body secretions, and significantly strengthen those arguments.”

We have added new data on ciliogenesis and on Notch signaling in suprabasal keratinocytes (Fig. S4), and data on the expression of various differentiation markers (Fig. S2D). Quantifications are provided in the corresponding text (page 11). To our surprise, these data indicate that ciliogenesis, Notch signaling and the differentiation program are largely normal in ninein-KO epidermis. Defects in differentiated cells are specifically seen at the level of protein abundance of selected markers, and at the level of desmosome assembly and lamellar body secretion.

“2. Along the same line of compounding effects, the authors make a point to discuss that the centrosomal function of Ninein are unaltered. As point of comparison, the authors refer to several mouse epithelial models of Plk4 overexpression, in which the observed phenotypes are different than the ones observed for the Ninein KO. This comparison is inaccurate, because Plk4 overexpression induces centrosome amplification rather than structural defects of the centrosome. Consequently, the phenotypes would be different as well. The more likely point of comparison for relating phenotypic changes would be mutations or ablations of centrosome appendage proteins, which would affect mother centriole activities. In that regards, defects in asymmetric cell divisions and ciliogenesis could further compound the phenotype and would again be an obvious culprit to investigate (see also point 1). Examination of asymmetric cell divisions should be more flushed out.”

In response to this suggestion, we studied the role of ninein at the centrosome and at the spindle in more detail. As pointed out to reviewers 1 + 2, we added an entire set of new data that demonstrate that spindle orientation is irregular in the absence of ninein, due to the loss of astral microtubules. We demonstrate spindle orientation defects both in cell cultures and in vivo, in the basal layer of the epidermis. We quantified the effects of ninein KO on the basal progenitor pool, on differentiation, on proliferation, mitotic index, and apoptosis. See new Figures 2C, Fig. 3, and supplementary Figures S3, and S4.

“3. The primary novelty of the study is the report of the first Ninein knockout mice. From that

standpoint, Figures 1 and 2, which utilize shRNA knockdown, do not significantly add to the studies. Could the authors not have conducted the microtubule organization studies using the knockout primary epidermal progenitor keratinocytes they established in culture, and provided the shRNA studies as support in supplemental figures?”

In the revised manuscript, we put more emphasis on ninein-KO in the epidermis, and we have removed a larger number of siRNA-based data from the main figures. We kept some data on the quantification of cortical microtubules in ninein-depleted cell cultures, since microtubule labelling in epidermal tissue has proven to be technically challenging (Lechler and Fuchs, 2007, JCB), and since we had established a reproducible protocol for in-vitro-differentiation of our cell cultures.

“It is also unclear how the tethering of the nuclei (Figure 2C-E) contribute to desmosomal defects.”

As pointed out to reviewers 1 + 2, we attempted to investigate a potential role of ninein in nuclear positioning during epidermal development in vivo, and we therefore compared nuclear positions in mouse tissue. Due to the small ratio of cytoplasm versus nuclear volume in suprabasal cell layers, we were technically unable to obtain solid information on centroid versus non-centroid nuclear positions. Since our previous experiments were exclusively obtained in cultured cells, and since the mechanisms that link nuclear positioning to the formation of cell adhesions in keratinocyte cultures and hair follicles were already investigated in a previous publication by Stewart et al (2015, J Cell Biol 209:403-18), we decided to drop them from this manuscript, except for a short comment on page 19.

“4. It is unclear whether the authors effectively achieve their desired Ninein knockout, because a more thorough characterization of the animals is missing from the figures. The authors report that the straight knockout animal (from PGK-Cre x Ninein fl/fl mating) are viable, with no further reporting of other phenotypic defects. This raises concern because two other groups have reported that Ninein is required for the development of neuronal lineages in vertebrate organisms (Dauber et al 2012, Wang et al 2009). If the animals are viable, are they born with Mendelian ratios? Do they have brain development defects? Do they display polydactyly? If they don't, is the entirety of the Ninein protein ablated in these animals?”

We now provide details on the ninein KO litters on page 6. Genotyping is reported in the new Fig. 1. Using qPCR with various primers, we tested the expression of multiple exons in the ninein KO mice in supplementary Fig. S1. Briefly, we have observed smaller litter sizes for the straight ninein KO, but the animals that survived show no visible defects in brain development. Moreover, we didn't see any defects that are otherwise seen in ciliogenesis-related mutant mice (such as renal defects or polydactyly). We think that the survivors were able to compensate the absence of ninein, whereas non-survivors were absorbed during early embryogenesis. Earlier reports on a role of ninein in brain development (Wang et al, 2009) may have come to different conclusions, since the timing of their experiments (using shRNA) did not allow for compensation of the ninein-loss. In our experiments, any early developmental defects that lead to lethality and any mechanisms of compensation of ninein-loss will of course be of future interest to us. Because of limitations of manpower, budget, and time, we weren't able to provide more detail at this point.

“The authors do provide a western blot and IF images of the loss of Ninein in Figure 3D/G. However, they use a C-terminal antibody to recognize the protein, so it is unclear which isoforms this antibody recognizes, and whether there is a truncated Ninein isoform remaining which alleviates some of their results. Could the authors use a larger panel of qPCR primer sets to detect Ninein and its isoform variants, spanning several exons junctions and, importantly, Exon1?”

Related to this, qPCR data have been added to the new supplementary Fig. S1A.

“5. The authors claim that the Ninein KO and cKO mice have flaky skin. The authors should provide evidence of this claim, in the form of an image of the flaky skin on the pups.”

We never use the term “flaky skin” in the manuscript. Nevertheless, we show a comparison of the skin from newborn mice from controls, cKO, and KO in supplementary Fig. S2A. Macroscopically, there are no obvious abnormalities in the skin from these different mice.

“6. The assays utilized to demonstrate defects in the differentiation of the epidermis (Figure 4) could be improved to be more informative. For example, the authors quantify the thickness of the epidermis based on H&E staining (Figure 4A/B), even though the stain itself looks different between the WT/Ninein-cKO and Ninein KO panels. It would be more informative if the authors quantified each compartment (basal, spinous, granular) based on compartment-specific markers (K5, K10, Involucrin).”

Immunostainings of K5, K10, and involucrin have been added to Fig. 2 C-D. The reduction of individual markers in ninein KO epidermis, such as involucrin, makes it difficult to compare layer thickness with controls. We therefore favored H&E staining, and indicated and quantified layer thickness in Fig. 2 A-B.

“The K1/K10 staining of the epidermis looks like it is staining the granular layers rather than the spinous layers (Figure S4A) and calls into question the validity of the antibody.”

We have substituted the keratin staining with new data, using new antibodies, now in Fig. 2C.

“Lastly, the authors should use EDU labeling counterstained with the K5 marker to understand the significance of the suprabasal proliferative cells in Figure 4C. If those cells are K5+, then they likely represent hyperproliferation of the basal layer, another attribute of ciliary dysfunction in the epidermis.”

We have added an experiment with EdU labeling to Fig. 3A. However, suprabasal EdU-positive cells are K10-positive. We provide corresponding text in the Results and Discussion sections for explanation.

Of minor concern:

“7. The text sometimes suffers from lack of clarity, grammatical errors, inaccurate diction, changing fonts.”

We have carefully revised the text, and we hope the the new manuscript will be easier to read.

“8. The legend and description of Figure 3C within the text are much too confusing to figure out and could benefit from simplification.”

We have altered the figure legend (now Fig. 1), as well as the corresponding text in the Results (pages 6/7), and we hope it will be easier to understand.

“9. The western blot in Figure 3G needs protein markers to delineate the size of the observed protein.”

This has now been added (new Fig. 1 F).

“10. In figure S3C, it is unclear what the arrows are pointing to. In the same figure legend, there is mention of a dotted line, but only a solid line is present on the image.”

The corresponding figure + legend have now been corrected.

“11. In figure S3D, the staining is suboptimal, making the interpretation of the p150-glued data challenging.”

We have added new, improved images of p150 (now new Fig. 4C).

Reviewer #1 Review

Comments to the Authors (Required):

Lecland and colleagues made substantial revisions to their first submitted manuscript and provide promising observations that could advance the understanding of skin development. However, this new version does not meet the quality standards of this journal. Not only the authors did not provide a mechanistic extension into how Ninein regulates nuclear positioning, but, and more importantly, they did not show or extend convincingly the mechanism through which Ninein regulates the formation of the cornfield layer. Furthermore, it would be fundamental that the authors substantially improve the quality of their images, as it is crucial to support the claimed observations. At this point it would be unhelpful to the scientific community to publish the revised manuscript.

Before submitting to a new journal the authors should consider addressing the following points:

- It is still unclear from this version why Ninein KO suprabasal cells are still cycling/dividing but the relative thickness of the basal and spinous layer is lower. If the authors are claiming that these cells must be transitioning faster to the subsequent granular and cornfield layers, they should perform lineage tracing experiments observing a putative higher number of labeled cells on those layers.

- Although the observation that Ninein is important to regulate spindle orientation in the basal layer and the cortical organization of microtubule, it remains unclear how its perturbation is contributing to the observed phenotypes (thinner epidermis and perturbed cornfield layer).

The quality of the image in Fig 2C is quite unacceptable. The images are blurry and/or not properly focused. The quality can be improved performing the acquisition either at the DeltaVision or the SP8 authors claim to use (described in the Materials and Methods section). Also, note that Ninein KO epidermis shows a smaller K10 layer although the authors say that there is no difference.

Filaggrin staining in fig 2D is not convincing, the image in the supplementary figure is more convincing.

Figure 3A - All the IF in sections throughout the manuscript should match the quality of this one.

Figure 3C - It is unclear why the authors are using MPEK with only one Ninein siRNA instead of using their Ninein KO isolated keratinocytes. Tubulin staining is over-saturated.

Figure 3D - It is unclear why the authors measure spindle orientation in cultured cells without inducing differentiation or using a more accurate model for studying asymmetric cell division. Furthermore it is unclear how the angles of division are being measured and what markers are being used. As it is, this experiment does not accurately add or support the importance of Ninein to properly orient mitotic spindle in the developing skin.

Figure 3F- It is very concerning to measure orientation angles from the images shown in Figure 3F. The stage of mitosis is hard to understand and guess (?) from the staining of only one marker.

Figure 4D-4F - The labels on the figure are not clear specially the x axis on the graphs

Figure 5A - The authors can use an inset in the top figure

Figure 6F - The authors should use a colored arrowhead as black arrowheads are hard to distinguish in the crowded cell

Reviewer #2 Review

Comments to the Authors (Required):

In this revised version, Lecland and colleagues have included new data strengthening the conclusions drawn from their results. In its present form, this manuscript provides a conceptual advance, since it gives insight into the functional roles of ninein in epidermal development.

Response to reviewer 1

1) “It is still unclear from this version why Ninein KO suprabasal cells are still cycling/dividing but the relative thickness of the basal and spinous layer is lower. If the authors are claiming that these cells must be transitioning faster to the subsequent granular and cornified layers, they should perform lineage tracing experiments observing a putative higher number of labeled cells on those layers.”

In fact, we see suprabasal cells that are EdU-positive, but no suprabasal cells undergoing division (page 9). Yet, epidermis in ninein-KO mice is thinner, probably due to altered cell morphology, unrelated to differences in the cell cycle (see answer to point 2). We think that defects in spindle orientation may affect the distribution of cell fate determinants in suprabasal cells, but consequences must be minor, since only 10% increase in Ki67-positive cells are detected, and since there is no change in the expression of early differentiation markers. These issues are discussed at the bottom of page 16 & top of page 17.

2) “Although the observation that Ninein is important to regulate spindle orientation in the basal layer and the cortical organization of microtubule, it remains unclear how its perturbation is contributing to the observed phenotypes (thinner epidermis and perturbed cornified layer).”

The thinner epidermis in ninein-KO mice must be due to altered cell morphology (cells are flatter, but the number of layers of suprabasal cells is comparable in ninein-knockouts and in controls). This is documented on page 8, and we propose that this may be due to the absence of a cortical microtubule network, which would be consistent with published data on ninein and cellular morphogenesis (see discussion on page 20).

We propose that alterations in the cornified layer are due to reduced secretion of lamellar bodies (Figure 6 and corresponding sections in Results and Discussion).

3) “The quality of the image in Fig 2C is quite unacceptable. The images are blurry and/or not properly focused. The quality can be improved performing the acquisition either at the DeltaVision or the SP8 authors claim to use (described in the Materials and Methods section). Also, note that Ninein KO epidermis shows a smaller K10 layer although the authors say that there is no difference.”

We have substituted the corresponding images by deconvolved ones (new Fig. 2C). These are now improved in contrast, and out-of-focus contribution has been eliminated. The thinner epidermis in ninein-KO is due to reduced cellular thickness (see response to point 2 and comment on page 8).

We don't claim that there is no difference between wt and ninein-KO, we only claim that the differentiation marker K10 is correctly expressed in suprabasal cells of both mice (page 8).

4) “Filaggrin staining in fig 2D is not convincing, the image in the supplementary figure is more convincing.”

The image in Figure 2D has been replaced. A clear difference in the amount of filaggrin is visible between wt and ninein-KO epidermis in the new Figure 2D. [This difference is even more visible when comparing wt and ninein-cKO (Fig. S2B), which may eventually reflect the potential of the full KO to functionally compensate for the loss of ninein, earlier in development (prior to the formation of the epidermis).]

5) “Figure 3C - It is unclear why the authors are using MPEK with only one Ninein siRNA instead of using their Ninein KO isolated keratinocytes. Tubulin staining is over-saturated.”

We also treated MPEK with a second Ninein siRNA and confirmed our results (data not shown). The second siRNA with the target sequence 5' GCAUUCUAAGCUACAAUGAtt3' (Ambion s70606) is listed in the revised Materials and Methods section (page 24).

The choice of MPEK over primary keratinocytes from WT and ninein-KO embryos was intentional, to have a homogenous pool of cells with a reproducible rate of symmetric division and unlimited propagation capacity.

Attempts of isolating primary keratinocytes resulted in a mixed pool of basal progenitor and suprabasal cells, displaying only very limited propagation capacity under cell culture conditions in our lab, and with the difficulty of dividing both parallel and perpendicularly (Lechler and Fuchs, 2005, Nature).

The tubulin immunofluorescence in Fig. 3C is overexposed deliberately, to show astral microtubules, which are the focus of this experiment. An explanatory comment has been added to the figure legend (page 41).

6) “Figure 3D - It is unclear why the authors measure spindle orientation in cultured cells without inducing differentiation or using a more accurate model for studying asymmetric cell division. Furthermore it is unclear how the angles of division are being measured and what markers are being used. As it is, this experiment does not accurately add or supports the importance of Ninein to properly orient mitotic spindle in the developing skin.”

As a perfect model for asymmetric divisions and differentiation, we have actually used epidermal sections from E16.5 embryos (documented in Figure 3F, G).

The cultures of MPEK (Fig. 3E) and HeLa (Fig. S3F) underline the general role of ninein in spindle orientation, a role that has not been described before.

The experimental approach (markers and determination of spindle angles) has been described in detail on page 26 (Materials and Methods).

7) “Figure 3F- It is very concerning to measure orientation angles from the images shown in Figure 3F. The stage of mitosis is hard to understand and guess (?) from the staining of only one marker.”

To verify that the cells are in mitosis, we have performed co-staining for phospho-histone H3 (H3P) and NuMA. We have added the panels of H3P staining to the revised Figure 3F. We admit that the exact phase of mitosis cannot be determined

accurately in this way. Nevertheless, from the high number of cells analyzed in epidermal sections we can clearly conclude that there is a shift from bimodal orientation (parallel/perpendicular) towards a more randomized and oblique orientation. We discuss the shortcomings of our approach briefly on page 10.

8) “Figure 4D-4F - The labels on the figure are not clear specially the x axis on the graphs”

We have improved the visibility of the labelling in the revised Figure 4E, G.

“Figure 5A - The authors can use an inset in the top figure”

The “insets” to Fig. 5A have been added directly below the low-mag figures (also see legend on pages 42/43).

“Figure 6F - The authors should use a colored arrowhead as black arrowheads are hard to distinguish in the crowded cell”

We added a white outline to the black arrows, to improve visibility (Figure 6 D, F).

March 11, 2019

RE: Life Science Alliance Manuscript #LSA-2019-00373-T

Dr. Andreas Merdes
University Toulouse III
Centre de Biologie du Developpement
118 route de Narbonne
Bat 4R3
Toulouse 31062
France

Dear Dr. Merdes,

Thank you for submitting your revised manuscript entitled "Epidermal development requires ninein for spindle orientation and cortical microtubule organization". Your manuscript was reviewed at another journal before, and the editors transferred those reports to us with your permission.

The reviewers who evaluated your work before expected further reaching mechanistic insight into the role of ninein in the epidermis. The reviewers also thought that some of the images shown were not of adequate quality and that the spindle measurements on skin sections were difficult to appreciate. Lack of mechanistic insight is not a concern for publication in Life Science Alliance and we invited you to address the other remaining concerns. We appreciate that you now provided a version that addresses the concerns regarding image quality and representation and that allows for a better understanding of how mitotic cells were analyzed in the skin sections. We also value the added discussion and are thus happy to accept your manuscript in principle for publication here.

Before sending you an official acceptance letter, please log into the system one more time and address the following:

- please upload all figure files as individual files
- note that the actin control in Fig 5B and 6B is the same, please indicate this in the legend in case Dsg1 and CDSN were analyzed on the same blot or replace with the correct control

A. FINAL FILES:

B. MANUSCRIPT ORGANIZATION AND FORMATTING:

Sincerely,

Andrea Leibfried, PhD

Executive Editor
Life Science Alliance
Meyershofstr. 1
69117 Heidelberg, Germany
t +49 6221 8891 502
e a.leibfried@life-science-alliance.org
www.life-science-alliance.org

March 12, 2019

RE: Life Science Alliance Manuscript #LSA-2019-00373-TR

Dr. Andreas Merdes
University Paul Sabatier/CNRS
Centre de Biologie du Developpement
118 route de Narbonne
Bat 4R3
Toulouse 31062
France

Dear Dr. Merdes,

Thank you for submitting your Research Article entitled "Epidermal development requires ninein for spindle orientation and cortical microtubule organization". It is a pleasure to let you know that your manuscript is now accepted for publication in Life Science Alliance. Congratulations on this interesting work.

*****IMPORTANT:** If you will be unreachable at any time, please provide us with the email address of an alternate author. Failure to respond to routine queries may lead to unavoidable delays in publication.*******

DISTRIBUTION OF MATERIALS:

Again, congratulations on a very nice paper. I hope you found the review process to be constructive and are pleased with how the manuscript was handled editorially. We look forward to future exciting

submissions from your lab.

Sincerely,
